# Maresin 1 repletion improves muscle regeneration after volumetric muscle loss

Jesus A Castor-Macias[1,2†], Jacqueline A Larouche[1,2†], Emily C Wallace[1], Bonnie D Spence[1], Alec Eames[1], Pamela Duran[1,2], Benjamin A Yang[1,2], Paula M Fraczek[1,2], Carol A Davis[3], Susan V Brooks[1,3], Krishna Rao Maddipati[4], James F Markworth[5], Carlos A Aguilar[1,2,6]*

[1]Department of Biomedical Engineering, University of Michigan, Ann Arbor, United States; [2]Biointerfaces Institute, University of Michigan, Ann Arbor, United States; [3]Department of Molecular & Integrative Physiology, University of Michigan, Ann Arbor, United States; [4]Department of Pathology, Lipidomics Core Facility, Wayne State University, Detroit, United States; [5]Department of Animal Sciences, Purdue University, West Lafayette, Indiana, United States; [6]Program in Cellular and Molecular Biology, University of Michigan, Ann Arbor, United States

**Abstract** The acute traumatic or surgical loss of skeletal muscle, known as volumetric muscle loss (VML), is a devastating type of injury that results in exacerbated and persistent inflammation followed by fibrosis. The mechanisms that mediate the magnitude and duration of the inflammatory response and ensuing fibrosis after VML remain understudied, and as such, the development of regenerative therapies has been limited. To address this need, we profiled how lipid mediators, which are potent regulators of the immune response after injury, varied with VML injuries that heal or result in fibrosis. We observed that non-healing VML injuries displayed increased pro-inflammatory eicosanoids and a lack of pro-resolving lipid mediators. Treatment of VML with a pro-resolving lipid mediator synthesized from docosahexaenoic acid, called Maresin 1, ameliorated fibrosis through reduction of neutrophils and macrophages and enhanced recovery of muscle strength. These results expand our knowledge of the dysregulated immune response that develops after VML and identify a novel immuno-regenerative therapeutic modality in Maresin 1.

*For correspondence:
caguilar@umich.edu

†These authors contributed equally to this work

Competing interest: The authors declare that no competing interests exist.

## Editor's evaluation

This is a compelling study on pro-inflammatory and anti-inflammatory lipids in relation to skeletal muscle injury. It convincingly identifies pro-inflammatory lipids during recovery predisposing to fibrosis, and maresin 1 as an anti-inflammatory lipid reducing fibrosis, improved muscle regeneration, partially restoring contractile function, of fundamental potential clinical applicagtion.

## Introduction

The acute loss of a large volume of skeletal muscle, or volumetric muscle loss (VML), is a severe type of injury that results in extensive fibrosis and loss of muscle function (*Garg et al., 2015b*; *Corona et al., 2015*). Despite attempts to use regenerative medicine and tissue engineering approaches (*Greising et al., 2019*), VML injuries typically result in sustained inflammation, supplantation of muscle tissue with excessive extracellular matrix (*Garg et al., 2015a*), and muscle fibrosis (*Greising et al., 2017*; *Aguilar et al., 2018*). The cellular and molecular drivers that recruit and program the fibrotic response (*Larouche et al., 2018*) after VML remain understudied. As such, regenerative therapies to restore muscle function after VML have found limited success.

VML results in incessant invasion and accumulation of inflammatory cells such as neutrophils and macrophages that normally protect the wound from infection (*Larouche et al., 2018*). The mobilization and infiltration of pro-inflammatory cells are critical both for host defense and the nascent stages of tissue repair (*Tidball, 2017*; *Wosczyna and Rando, 2018*). However, VML drives a dysregulated immune cascade, resulting in increased and prolonged inflammation (*Larouche et al., 2023*), inadequate removal of debris, adoption of a fibrotic, TGFβ1-secreting phenotype in macrophages (*Chowdary et al., 2023*; *Larouche et al., 2022*), and aberrant regenerative dynamics of tissue-resident stem cells (*Larouche et al., 2022*; *Stepien et al., 2020*). Yet, the causative extracellular signaling factors that recruit and condition the pathological immune cell responses have been underexamined (*Tidball, 2005*; *Novak et al., 2014*; *Wang et al., 2014*).

Recently, a new class of bioactive signaling factors derived from omega-3 or essential fatty acids called pro-resolving lipid mediators (*Serhan, 2014*) have been discovered to regulate the magnitude and duration of the inflammatory response (*Serhan et al., 2008*). These pro-resolving lipid mediators have been shown to restrain the infiltration of neutrophils (*Malawista et al., 2008*), augment macrophage polarization (*Schmid et al., 2016*) and phagocytosis (*Serhan et al., 2012*), and attenuate pro-inflammatory signaling molecules, such as eicosanoids (*Serhan and Chiang, 2013*). The balance that develops after VML between classical inflammatory eicosanoids derived from arachidonic acid (e.g., prostaglandins and leukotrienes) (*Wang et al., 2021*; *Dennis and Norris, 2015*) and pro-resolving lipid mediators such as resolvins, protectins, and maresins has not been evaluated. Moreover, how many of these lipid mediators contribute to recruitment of immune cell subtypes after VML and concomitant fibrosis requires further understanding.

Herein, we contrasted VML injuries that heal and restore muscle function with those that result in fibrosis and loss of muscle function. We used metabolipidomics analysis over a time course to assess the changes in the composition of bioactive signaling mediators for VML injuries that regenerate compared to those that result in fibrosis. For degenerative VML injuries, increased pro-inflammatory eicosanoids were detected when compared to VML injuries that heal and no detectable change in pro-resolving mediators. Exogenous administration of a docosahexaenoic acid-derived pro-resolving lipid mediator called Maresin 1 (*Serhan et al., 2012*; *Marcon et al., 2013*) after degenerative VML injury was observed to impact resolution trajectory by simultaneously attenuating macrophage and neutrophil infiltration, reducing fibrosis, and promoting muscle regeneration via enhancing muscle stem cell (MuSC) proliferation. These findings suggest pro-resolving lipid mediators can encourage healing of severe muscle trauma and alter the signaling environment to support MuSC-based regeneration.

## Results

### Comparative analysis of VML injuries of varied sizes reveals variations in fibrosis and function

To establish a framework for understanding how variations in the inflammatory response drive fibrotic scarring and muscle degeneration after VML injury, we administered bilateral VML injuries to the tibialis anterior (TA) muscles of adult C57BL/6J mice by delivering full-thickness 1-mm or 2-mm punch biopsies (*Anderson et al., 2019*; *Figure 1A*). We extracted muscles at 7 and 14 d post injury (dpi) and observed increased collagen deposition in 2-mm defects when compared to 1-mm defects by Picrosirius red staining (*Figure 1B–D*, n = 3–4 mice per condition, paired). To determine whether the increases in fibrosis with larger VML defects engendered reductions in maximal tetanic force, we compared 1-mm and 2-mm defects with uninjured muscle at 28 dpi. We found reductions in force output for 2-mm defects when compared to 1-mm and uninjured muscle (*Figure 1E and F*, *Figure 1— figure supplement 1A–E*, n = 6–8 mice per group, unpaired two-way ANOVA), which is consistent with previous studies (*Larouche et al., 2022*; *Anderson et al., 2019*). Summing these results shows that 2-mm punch biopsy defects to murine TA muscles (degenerative VML injuries) produce fibrotic supplantation and reductions in muscle function, while 1-mm punch biopsy defects (regenerative VML injuries) result in less fibrosis and functionally recover to the same level as uninjured tissues by 28 dpi.

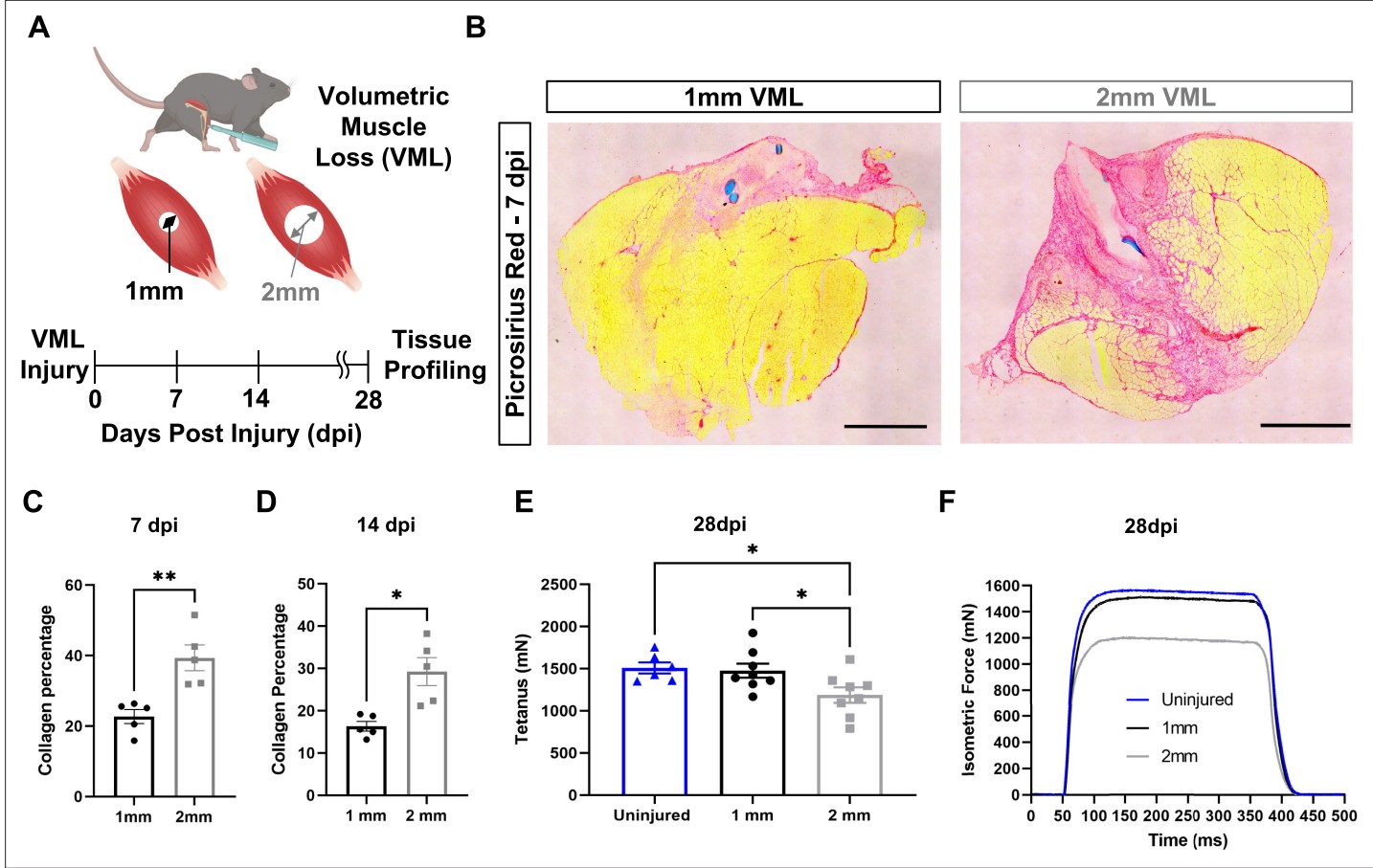

**Figure 1.** In vivo comparison of degenerative and regenerative volumetric muscle loss (VML) injuries. (**A**) Schematic of experiment, whereby young (3 mo) mice were administered 1-mm (administered to left leg) or 2-mm (administered to right leg) biopsy punches to their tibialis anterior muscle to create a VML injury. (**B**) Representative cross-sections stained with Picrosirius red isolated from healing (1-mm defect) and non-healing (2-mm defect) 7 d post injury (dpi) (n = 4–5 tissues from five mice per group). Scale = 200 µm. (**C, D**) Quantitation of images from (**B**) shows reductions in Picrosirius red for 1-mm defects compared to 2-mm defects at 7 dpi (**C**) and 14 dpi (**D**). Graphs show mean ± SEM. (**E**) Average tetanic force from uninjured (blue) tibialis anterior muscle at 28 dpi following 1-mm (black) or 2-mm (gray) VML injuries. Bars show mean ± SEM and *p<0.05 between injury types by two-way ANOVA and post hoc. n = 6–8 tissues 6–8 mice per group. (**F**) Representative force curves of uninjured tibialis anterior muscle (blue) at 28 dpi following 1-mm (black) or 2-mm (gray) injuries. For (**C**) and (**D**), unpaired *t*-test with Welch's correction. *p<0.05 and **p<0.01.

The online version of this article includes the following figure supplement(s) for figure 1:

**Figure supplement 1.** Functional assessment of response to healing or degenerative volumetric muscle loss injuries.

## Metabolipidomic profiling after volumetric muscle injuries shows imbalances in pro- and anti-inflammatory lipid mediators

A dysregulated immune response (*Larouche et al., 2018*; *Larouche et al., 2022*) has been shown to be responsible for the fibrotic scarring induced from degenerative VML injury. To glean the factors that recruit and program myeloid-derived cells to promote excessive tissue fibrosis, we administered regenerative (1 mm) vs degenerative (2 mm) VML injuries to TA muscles as above and performed liquid chromatography coupled to tandem mass spectrometry (LC-MS/MS)-based metabolipidomics profiling at 0, 3, 7, and 14 dpi (*Figure 2A*). We profiled a total of 143 lipid mediator species across the time course of recovery from VML. In total, 80 lipid mediators were reliably detected in muscle tissue homogenates (signal-to-noise ratio >3 and peak quality > 0.2 in at least 50% of samples, *Supplementary file 1* ).

Of the 80 analytes detected by LC-MS/MS, 53 displayed differential dynamics between regenerative vs degenerative VML injuries. For degenerative defects, a greater abundance of many pro-inflammatory eicosanoids such as the major lipoxygenase-derived hydroxyeicosatetraenoic acids (HETEs) 5-HETE, and 12-HETE, as well as cyclooxygenase (COX)-derived prostaglandins, including

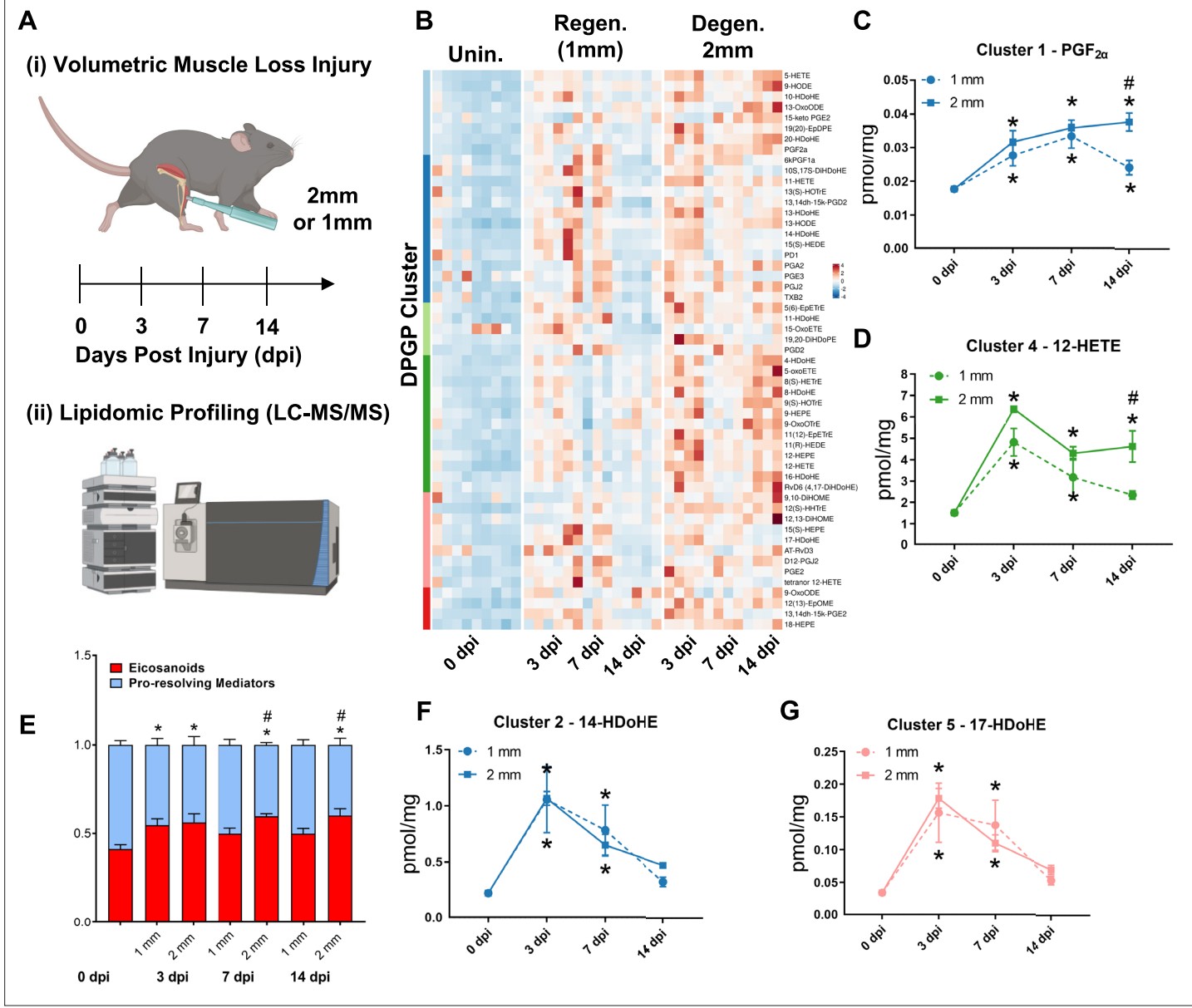

**Figure 2.** Lipidomic profiling after volumetric muscle loss (VML) injuries reveals stronger and sustained lipid mediator response in degenerative injuries. (**A**) Mice were administered a bilateral defect to tibialis anteriors (TAs) (left leg received a 1-mm defect and right leg received a 2-mm defect). Tissues were harvested at 0, 3, 7, and 14 d post injury (dpi) and subjected to liquid chromatography coupled to tandem mass spectrometry (LC-MS/MS) analysis. n = 5 tissues from five mice per injury type and timepoint. Statistical analyses were paired. (**B**) Row-scaled heatmap of 53 differentially expressed analytes across injuries and timepoints. Analytes are grouped by Dirichlet Process Gaussian Process (DPGP) clusters. (**C–D, F–G**) Changes of specific lipid metabolites in different clusters per injury type and timepoint expressed in pmol/mg. Two-way paired ANOVA was used to estimate statistical differences between injury and timepoints. Graphs show mean ± SEM, *<0.05 between timepoints of matched injury timepoint, **<0.01 between different injury types, and #<0.05 between different injury types at a certain timepoint. (**C, D**) Prostaglandin F2 alpha and 12-HETE are both eicosanoids related to pro-inflammatory effects. (**E**) Eicosanoids (TXB$_2$, PGD$_2$, PGE$_2$, PGF$_{2\alpha}$, 6kPGF$_{1\alpha}$, and 5-, 12-, 15-HETEs) vs pro-resolving mediators pathway markers (5-HEPE, 4-HDoHE, 7-HDoHE, 12-HEPE, 14-HDoHE, RvD6, PD1, 10S,17S-DiHDoHE, Maresin 1, Mar1-n3DPA, LXA$_4$) graphed for 2-mm and 1-mm VML defects at 0, 3, 7, and 14 dpi. Both analyte groups were normalized to ratios adding up to 1. Two-way paired ANOVA statistical tests were used to compare values between injury and timepoints. *p<0.05 between timepoints of matched injury timepoint, and #p<0.05 between different injury types at a certain timepoint. (**F**) 14-HDoHE represents a pathway marker for maresins. (**G**) 17-HDoHE is a pathway marker for D-resolvins/protectins. Both maresins and resolvins are related to anti-inflammatory effects.

The online version of this article includes the following figure supplement(s) for figure 2:

**Figure supplement 1.** Dynamics of lipid mediators after volumetric muscle loss injury.

$PGF_{2\alpha}$, $PGE_2$, $PGD_2$, $PGA_2$, $PGI_2$ (6-keto-$PGF_{1\alpha}$), and $TXB_2$, was detected (**Figure 2B**). To gain further insights into time-series variations between the two injuries, we performed non-parametric clustering of differentially detected analytes (**McDowell et al., 2018a**; **Figure 2—figure supplement 1A**). We observed variations in temporal profiles for classical eicosanoids such as prostaglandin ($PGF_{2\alpha}$, cluster 1) and 12-HETE (cluster 4), whereby for regenerative defects, a temporal increase in concentration was detected followed by returns to uninjured levels by 14 dpi. In contrast, $PGF_{2\alpha}$ and 12-HETE remained elevated in concentration for degenerative defects at 7 and 14 dpi (**Figure 2C and D**). In further agreement with this observation, parametric statistical analysis of analytes pooled by specific biosynthetic pathways that lead to pro-inflammatory eicosanoid production such as 5-LOX and 12-LOX showed increases in concentration for degenerative VML injuries at longer timepoints driven mainly by greater and/or more prolonged local biosynthesis of $PGF_{2\alpha}$ (COX pathway), 5-HETE (5-LOX pathway), and 12-HETE (12-LOX pathway) (**Figure 2—figure supplement 1B–F**).

To determine whether the increase in eicosanoids for degenerative defects was balanced by increases in specialized pro-resolving mediators and their related pathway markers/biosynthetic intermediates (5-HEPE, 4-HDoHE, 7-HDoHE, 12-HEPE, 14-HDoHE, RvD6, PD1, 10S, 17S-DiHDoHE, $Mar1_{n3DPA}$, and $LXA_4$), we plotted the ratio between classical pro-inflammatory eicosanoids (sum of $TXB_2$, $PGD_2$, $PGE_2$, $PGF_{2\alpha}$, $6kPGF_{1\alpha}$, and 5-, 12-, 15-HETEs) relative to detected specialized pro-resolving mediators and their related pathway markers (**Figure 2E**). These data revealed at 14 dpi an overall stronger inflammatory response for degenerative defects when compared to regenerative defects. In contrast to the increased and sustained levels of eicosanoids, pro-resolving pathway markers/biosynthetic intermediates such as from Maresin 1 (14-HDoHE) and D-series resolvins/protectins (17-HDoHE) from clusters 2 and 5 transiently increased in abundance and returned to uninjured levels by 14 dpi (**Figure 2F and G**). The pro-resolving mediators also displayed a highly similar trajectory between degenerative or regenerative injuries (**Figure 2F and G**). Overall, the distinct intramuscular lipid mediator profile between injury types and timepoints suggests a dysregulated immune response may be driven, in part, by a relative overabundance of classical pro-inflammatory eicosanoids within degenerative VML injuries in the absence of a coordinated pro-resolving lipid mediator response.

## Treatment of VML with Maresin 1 reduces fibrosis and inflammatory cell accumulation

Our data suggest that the lack of pro-resolving mediators and lipid mediator imbalances may contribute to a lack of tissue regeneration after VML. To examine if treatment of degenerative VML with a pro-resolving lipid mediator reduced inflammation and fibrosis, we administered synthetic Maresin 1 (7R,14S-dihydroxydocosa-4Z,8E,10E,12Z,16Z,19Z-hexaenoic acid) (**Serhan et al., 2012**), which has previously been shown to reduce neutrophil accumulation and alter macrophage phenotype during tissue regeneration (**Serhan et al., 2012**; **Marcon et al., 2013**). Following degenerative VML injury, we locally administered synthetic Maresin 1 through intramuscular injection every other day beginning at 1 dpi (**Figure 3A**). At 7 dpi, a significant reduction in collagen deposition was observed by Picrosirius red staining for muscles treated with Maresin 1 compared to vehicle-treated contralateral limbs (**Figure 3B and C**, n = 9 tissues from nine mice, paired). Based on observed reductions in collagen deposition, we next sought to understand the differences in inflammatory cell abundance. Both immunohistochemistry stains for CD68 and flow cytometry quantifications (CD45$^+$F4/80$^+$) revealed a reduction for muscles treated with Maresin 1 compared to vehicle-treated controls (**Figure 3D and E**), n = 7 tissues from seven mice, paired; (**Figure 3—figure supplement 1A and B**, n = 8–10 muscles from five mice, unpaired). Moreover, consistent with literature showing reduced neutrophil accumulation as a result of Maresin 1 treatment (**Francos-Quijorna et al., 2017**), flow cytometry at 7 dpi for CD45$^+$Ly6G$^+$ cells revealed significant reductions (**Figure 3—figure supplement 1A–D**, n = 8–10 tissues from five mice, unpaired). These results suggest that administration of Maresin 1 after VML suppresses inflammation and concomitant fibrosis.

## Administration of Maresin 1 after VML augments muscle force

To determine whether the reductions in inflammation and fibrosis from Maresin 1 treatment improved muscle regeneration and restoration of muscle force, we utilized a MuSC fluorescent reporter model (Pax7$^{CreERT2}$ – Rosa26$^{-TdTomato}$). Upon administration of tamoxifen and Cre recombination, this model indelibly labels MuSCs and their progeny with TdTomato. We injured TA muscles with 2-mm VML as

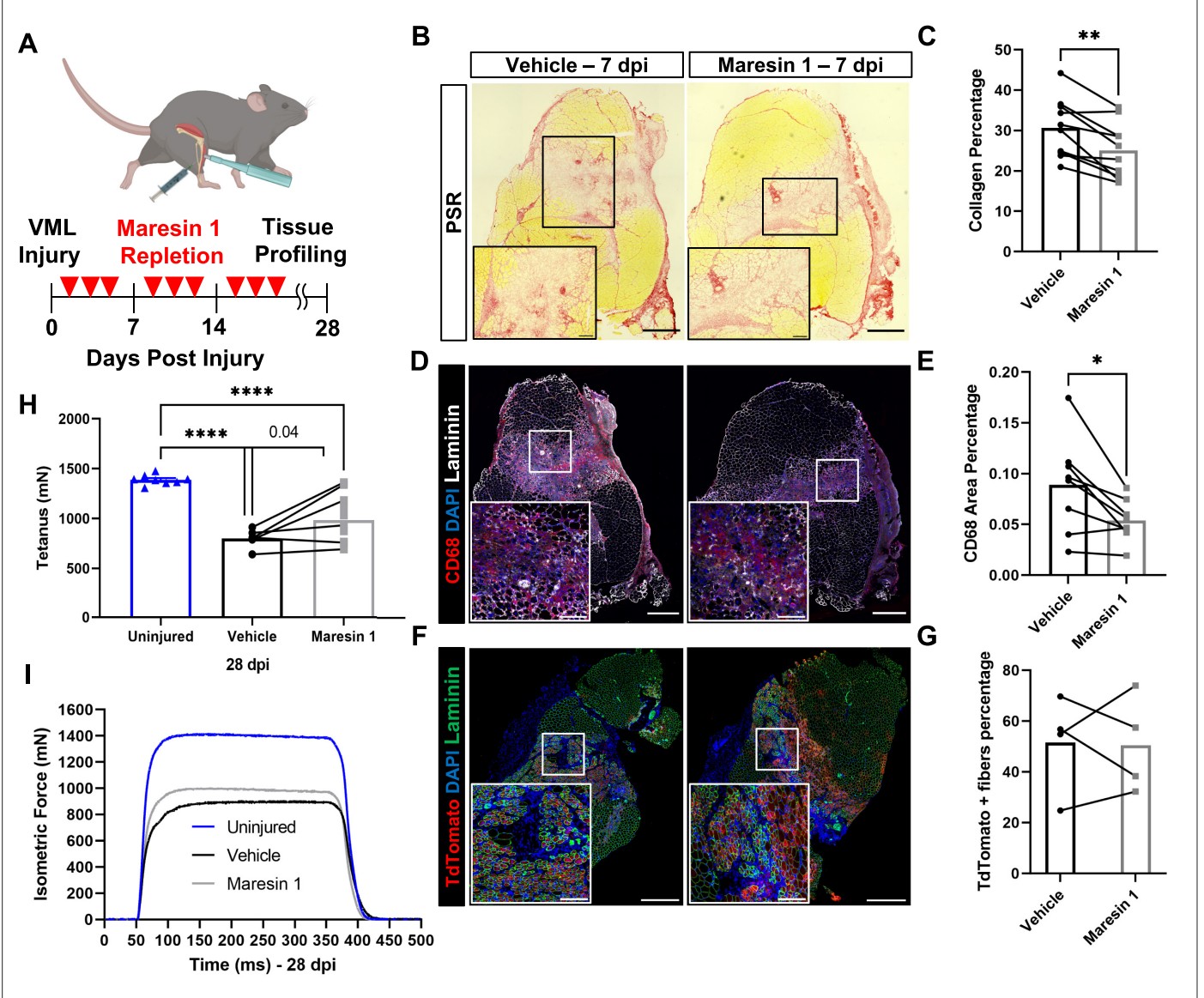

**Figure 3.** Repletion of Maresin 1 after degenerative volumetric muscle loss (VML) injury results in reductions in inflammation and fibrosis. (**A**) Mice were administered bilateral 2-mm VML defects to their tibialis anterior (TA) muscles. One limb received an intramuscular injection of Maresin 1 and the other limb received vehicle (saline + 0.01% EtOH) treatment every 2 d. (**B**) Representative cross-sections stained with Picrosirius red isolated 7 d post injury (dpi) following treatment with vehicle or Maresin 1 treatment. n = 7 tissues from seven mice per group, scale bar = 500 μm. Inset scale bar = 150 um. *p<0.05 by paired t-test.(**C**) Quantitation of Picrosirius red at 7 dpi from (**B**) shows a reduction of collagen deposition for Maresin 1 treatment. Graphs show mean ± standard error of mean (SEM), **p<0.01 by paired t-test. (**D**) Representative cross-sections from muscles isolated 7 dpi treated with vehicle or Maresin 1 treatment. Sections are immunostained for CD68 (red), DAPI (blue), and laminin (white). n = 7 tissues from seven mice per group, scale bar = 500 μm. Inset scale bar = 150 um. (**E**) Quantitation of number of macrophages (CD68⁺) at 7 dpi from (**D**) shows a decrease in the number of macrophages for Maresin 1 treatment. Graphs show mean ± SEM, *p<0.05 by paired t-test. (**F**) Representative cross-sections from muscles isolated 7 dpi treated with vehicle or Maresin 1 treatment. Sections are immunostained for TdTomato (red), DAPI (blue), and laminin (green). Scale bar = 500 μm. Inset scale bar = 150 um. (**G**) Quantitation of average cross-sectional area of fibers positive for TdTomato. n = 4 tissues from four mice per group. (**H**) Average tetanic force from muscle stimulation of uninjured (blue) TA muscle at 28 dpi and treatment with saline (black) and Maresin 1 (gray). Bars show mean ± SEM and ****p<0.0001 between uninjured and VML + Maresin 1 treatment, and between uninjured and VML + saline treatment, and p=0.0477 between VML + Maresin 1 and VML + saline treatment by one-way ANOVA and post hoc. n = 11 tissues from 11 mice per group. (**I**) Representative force curves of uninjured TA muscle (blue) at 28 dpi following saline (black) and Maresin 1 treatment (gray).

The online version of this article includes the following figure supplement(s) for figure 3:

**Figure supplement 1.** Immune profiling of response to volumetric muscle loss (VML) injury after treatment with Maresin 1.

**Figure supplement 2.** Regenerative response to volumetric muscle loss (VML) injury after treatment with Maresin 1.

**Figure supplement 3.** Changes in muscle force after volumetric muscle loss (VML) injury are enacted with treatment with Maresin 1.

above, extracted muscles at 7 dpi, and cross-sectioned and immunostained the tissues for TdTomato (*Larouche et al., 2021*). We observed an increase in the cross-sectional area (CSA) of TdTomato⁺ myofibers for tissues treated with Maresin 1 when compared to untreated muscles (*Figure 3F and G*, *Figure 3—figure supplement 2D–E*, n = 4 tissues from four mice, paired). We further validated these results by immunostaining for myosin heavy chain 3 (MyHC3) at 7 dpi. In line with our previous observations with the MuSC lineage-tracing model, we detected a small shift in the distribution of regenerating myofibers, with slightly larger MYH3⁺ myofibers for tissues treated with Maresin 1 when compared to vehicle-treated tissues (*Figure 3—figure supplement 2A–C*, n = 11 tissues from 11 mice, unpaired), respectively. To glean whether reductions in fibrosis and small increases in muscle regeneration resulted in changes in recovery of muscle force, we measured maximal tetanic force at 28 dpi for Maresin 1-treated tissues and vehicle-treated controls. We detected treatment of VML with Maresin 1 yielded increases in maximal tetanic force when compared to vehicle alone (*Figure 3H and I*, *Figure 3—figure supplement 3A–E*, n = 11 tissues from 11 mice, unpaired). These results suggest that repletion of Maresin 1 positively promotes restoration of function through reduction of inflammation and fibrosis and slight increases in muscle regeneration.

## Maresin 1 engagement of LGR6 receptor mediates proliferation in MuSCs

Previous research (*Chiang et al., 2019*) has demonstrated that Maresin 1 selectively binds to the lleucine-rich repeat containing G-protein-coupled receptor 6 (LGR6), but not other LGR receptors, such as LGR4 or LGR5. To understand whether LGR6 is expressed by MuSCs and has direct impact across different MuSC states (quiescent, activated, differentiated), we assessed changes in *Lgr6* expression via RT-qPCR at three timepoints (immediately post MuSC isolation from uninjured limb muscles, following in vitro activation and culture, and 72 hr post differentiation induction using low-serum media). We isolated MuSCs from uninjured hindlimb muscles (*Aguilar et al., 2016*; *Yang et al., 2021*) and observed low but detectable expression of *Lgr6* among freshly isolated MuSCs and differentiated, fused myotubes, with a nearly 65-fold increase in expression among proliferating myoblasts (*Figure 4A*, n = 3 wells per condition). This is consistent with previously published bulk RNA-seq datasets isolated from MuSCs after barium chloride injury showing upregulated *Lgr6* expression among MuSCs that were isolated after injury (*Shcherbina et al., 2020*), and suggests that activation of the LGR6 receptor by Maresin 1 may influence proliferation (*Khedgikar et al., 2022*). To test whether Maresin 1 stimulated proliferation, we isolated MuSCs from uninjured limb muscles and exposed them to Maresin 1 in the presence of 5-ethynyl-2'deoxyuridine (EdU) for 24 hr. In line with our hypothesis, we observed a significant increase in EdU-positive cells as a result of Maresin 1 treatment (*Figure 4B*, *Figure 4—figure supplement 1A*, n = 4 wells per condition, unpaired).

## Maresin 1 promotes cyclic AMP and not canonical Wnt in MuSCs

Engagement of LGR6 in different cell types has been shown to stimulate several different pathways, including cyclic AMP (cAMP) (*Khedgikar et al., 2022*) and canonical Wnt signaling (*Khedgikar et al., 2022*; *Liu et al., 2019*), which have been associated with MuSC proliferation, migration, and differentiation (*Berdeaux and Stewart, 2012*; *Yin et al., 2013*). To examine whether Maresin 1 can promote G-protein-coupled receptor (GPCR) signaling and cyclic AMP, we isolated and cultured MuSCs with Maresin 1 and evaluated cAMP levels. In line with our previous observations, MuSCs treated with Maresin 1 exhibited increased cAMP compared with controls (*Figure 4C*, n = 8 biological replicates [three wells per replicate], paired). Engagement of LGR6 has also been demonstrated to influence canonical Wnt signaling (*Khedgikar et al., 2022*; *Liu et al., 2019*), and to examine this effect we developed a *Pax7Cre^{ERT2}-Rosa26^{Tcf/Lef-LSL-H2B-GFP}* reporter mouse (P7Wnt) that activates a nuclear green fluorescent protein reporter in MuSCs in response to canonical Wnt signaling (*Figure 4D*). We validated this mouse model by culturing MuSCs harvested from uninjured P7Wnt hindlimb muscles with recombinant Wnt3a protein for 48 hr and immunostaining for GFP. As expected, we detected an increased percentage of GFP⁺ cells relative to vehicle treatment (*Figure 4—figure supplement 1B*). We next isolated MuSCs from uninjured P7Wnt hindlimb muscles, cultured the cells with Maresin 1, and observed no change in GFP/canonical Wnt signaling compared to controls (*Figure 4E*, n = 6 wells per condition, unpaired). Combining these results suggests that Maresin 1 stimulates MuSC proliferation via cAMP signaling, and this effect is not mediated through canonical Wnt, which is consistent with

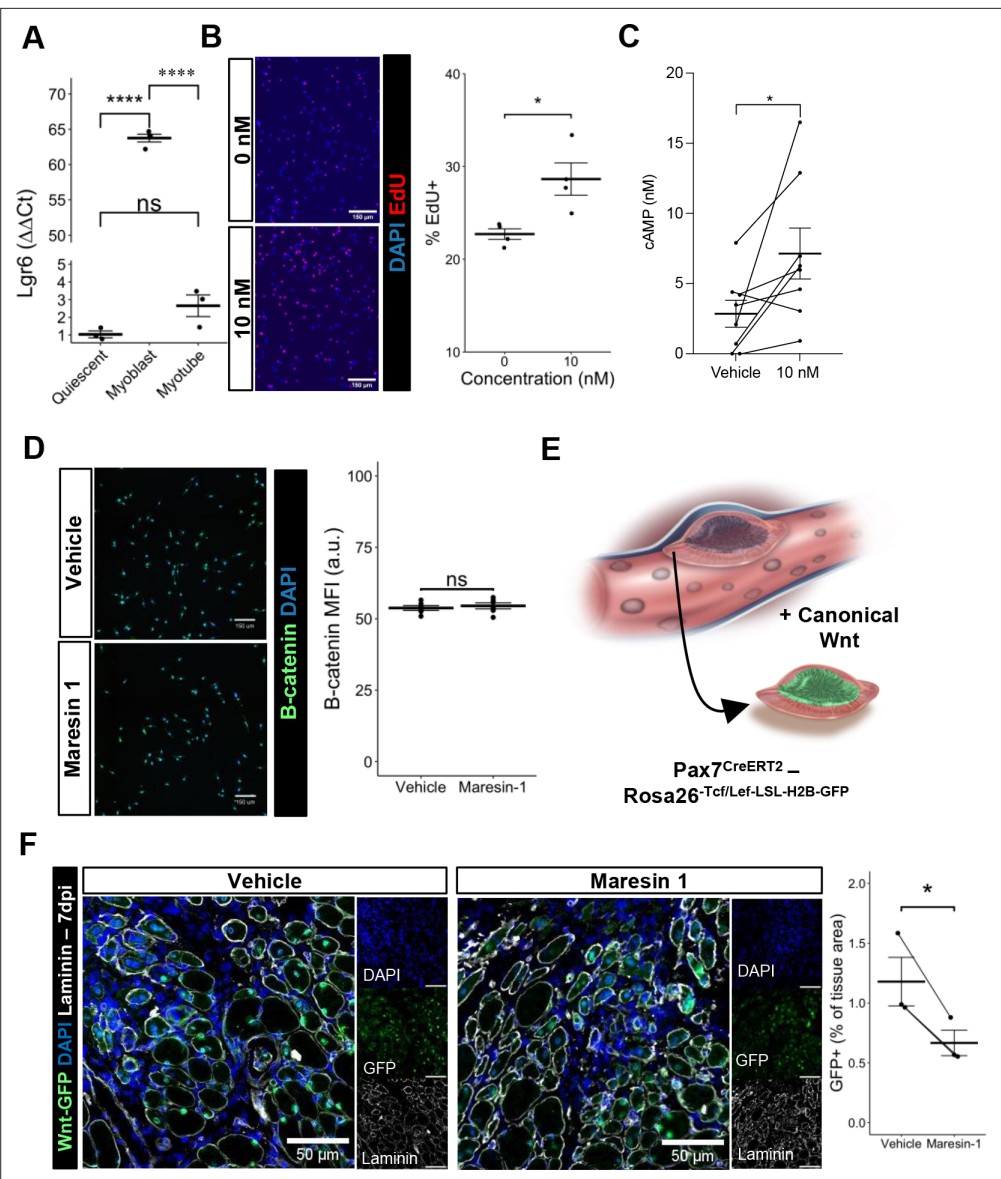

**Figure 4.** Maresin 1 impacts muscle stem cell (MuSC) proliferation through Lgr6. (**A**) Lgr6 is highly upregulated in proliferating myoblasts by RT-qPCR. Graph shows mean ± SEM, ****$p<0.0001$ by one-way ANOVA and Benjamini–Hochberg (BH) post hoc analysis. n = 3–4 wells per condition. (**B**) Treatment of myoblasts with a low concentration of Maresin 1 in vitro increases proliferation based on EdU incorporation over 24 hr. Graph shows mean ± SEM, *$p<0.05$ by one-way ANOVA with BH post hoc analysis. n = 4 wells per condition, where each well was quantified using four separate 10× images. Scale bars indicate 150 um. (**C**) Measurement of cyclic AMP in primary myoblasts treated with 10 nM Maresin 1 or vehicle for 24 hr. n = 8 biological replicates, each value was calculated by averaging three wells per condition for each mouse. Data presented as mean ± SEM. *$p<0.05$ by two-sided paired *t*-test. (**D**) β-Catenin levels are unchanged following Maresin 1 treatment of wildtype MuSCs in vitro. ns denotes $p>0.05$ by two-sided, two-sample *t*-test. n = 6 wells per condition. (**E**) Schematic of MuSC lineage reporter for canonical Wnt signaling. In response to canonical Wnt, cells express green fluorescent protein in their nuclei. (**F**) In vivo treatment of volumetric muscle loss (VML) injuries with Maresin 1 reduces activation of canonical Wnt signaling in Pax7+ MuSCs using the P7Wnt-GFP mouse model at 7 d post injury (dpi). Graph shows mean ± SEM, *$p<0.05$ by two-sided, two-sample *t*-test. n = 3 mice per condition. Scale bars indicate 100 um.

The online version of this article includes the following figure supplement(s) for figure 4:

**Figure supplement 1.** Experimental evaluation of parameters for Maresin 1 treatment of muscle stem cells (MuSCs) and their progeny in vitro.

previous results (*Khedgikar et al., 2022*; *Ruan et al., 2019*). To determine whether Maresin 1 treatment altered canonical Wnt in MuSCs in vivo after degenerative VML injuries, we injured P7Wnt mice with bilateral degenerative VML and treated one limb with vehicle and the other limb with Maresin 1. In vivo treatment of degenerative VML injuries with Maresin 1 showed reductions in GFP and canonical Wnt signaling among MuSCs at 7 dpi (*Figure 4F*, n = 3–4 tissues from three mice, unpaired). Given sustained Complement activates canonical Wnt signaling and muscle fibrosis (*Larouche et al., 2018*; *Naito et al., 2012*), this result was in line with our observations of reductions in collagen deposition and macrophage accumulation. We further investigated why stronger enhancements in muscle regeneration were not further enhanced given activation of cAMP and reductions in canonical Wnt by focusing on TGFβ1, which has been shown to attenuate MuSC fusion and becomes dysregulated after VML (*Larouche et al., 2022*; *Girardi et al., 2021*). We tested whether changes in active TGFβ1 were derived with Maresin 1 treatment after VML using ELISA. We observed slight reductions in active TGFβ1 at 7 dpi in VML-injured muscles treated with Maresin 1 compared to those treated with vehicle (*Figure 4—figure supplement 1C*, n = 3–4 tissues from 3 to 4 mice, unpaired). Integrating these results supports the notion that Maresin 1 positively impacts MuSC proliferation through cAMP, but fusion of MuSCs and associated myofiber repair is not significantly enhanced in vivo.

## Single-cell RNA sequencing supports reductions in inflammatory cells and increases in myogenic cells as a result of Maresin 1 administration

To further probe the impact of Maresin 1 treatment post VML, we performed droplet-based single-cell RNA sequencing (scRNA-seq) on viable mononucleated cells isolated from vehicle and Maresin 1-treated degenerative VML defects at 7 dpi (*Figure 5A*, each condition represents a pool of four tissues from four mice). We generated 21,679 high-quality scRNA-seq libraries from the Maresin 1-treated and vehicle-treated tissues, respectively, encompassing on average 2592 genes per cell with an average read depth of 10,873 unique molecular identifiers (UMIs) per cell (*Figure 5—figure supplement 1A*). Each dataset was log normalized before being integrated using Seurat v4, followed by principal component analysis (PCA), unsupervised Louvain clustering, and Uniform Manifold Approximation and Projection (*Shcherbina et al., 2020*) to reveal 13 cell types (*Figure 5A and B*). Cluster-based cell-type annotation was performed according to the expression of known marker genes (*Figure 5—figure supplement 1B*) and alignment with previously published datasets (*Chowdary et al., 2023*). Consistent with immunohistological stains and flow cytometry, we observed reductions in neutrophils and dendritic cells in addition to increases in endothelial cells and MuSCs for VML-injured tissues treated with Maresin 1 (*Figure 5C*). MAST differential expression testing further revealed substantially different transcriptional landscapes within cell types across treatments, especially among immune (macrophages, neutrophils, T cells) and myogenic (myonuclei, MuSCs) cells (*Figure 5D*). Differential gene expression in macrophages treated with Maresin 1 revealed downregulation of inflammatory signaling (*Ly6c2, Hmox1, Ccl2*) and genes associated with fibrosis (*Thbs1, Tgfbi, Mmp9*), consistent with our histological observations (*Figure 5E*). Among MuSCs, the predominant gene sets reduced with Maresin 1 treatment were inflammatory (Ccl4, Il1b), including several transcripts related to interferon signaling response (Cxcl10, Cxcl9). Upregulated markers in MuSCs included those associated with myogenic differentiation (Mymk, Myog, Ttn) (*Figure 5F*). Together, these results support the regenerative impact of Maresin 1 treatment being realized principally through both promoting the expansion and differentiation of myoblasts and reducing immune-cell-induced inflammation and fibrosis.

## Discussion

VML is a devastating type of acute trauma that results in fibrosis and loss of muscle function. Inadequate understanding of the drivers of these pathological outcomes has reduced the efficacy of many different types of regenerative therapies (*Greising et al., 2019*), and, as such, VML repair remains an unmet clinical need. Herein, we established a critical threshold model for VML in murine TA muscles. Consistent with previous observations in an analogous model in the rectus femoris (*Larouche et al., 2022*; *Anderson et al., 2019*), VML defects below a critical size (1 mm) regenerate, while larger VML defects (2 mm) result in long-term increases in fibrosis and reductions in contractile force. Using this model of regenerative and degenerative VML, we performed metabolipidomics analysis (*Markworth*

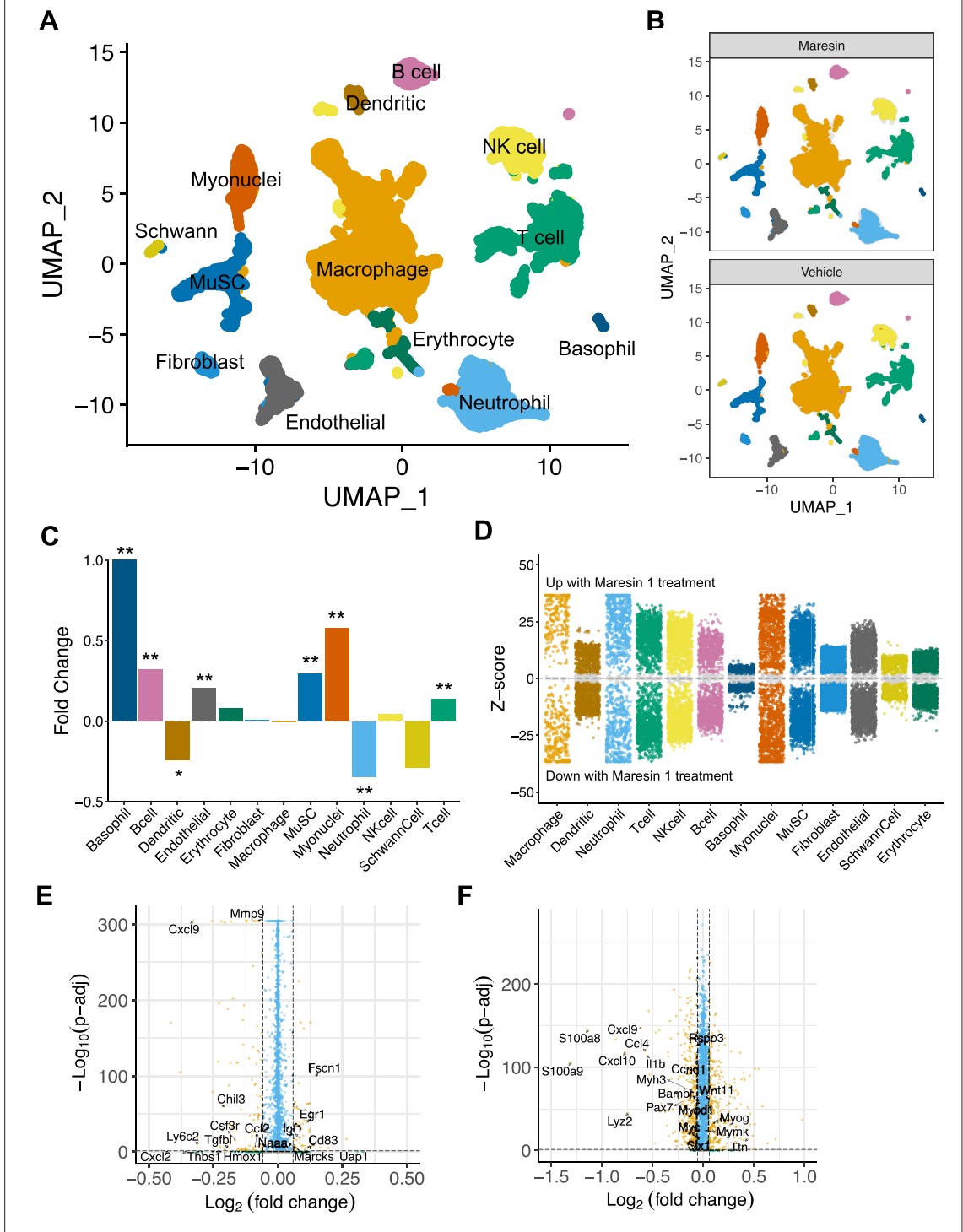

**Figure 5.** Cellular and molecular response to Maresin 1 repletion after degenerative volumetric muscle loss supports enhanced myogenic response and reduction of fibrotic macrophages at 7 d post injury (dpi). (**A**) Uniform Manifold Approximation and Projection (UMAP) plots colored by clusters at a resolution of 0.2 and annotated into cell types. (**B**) UMAP plots split by treatment and colored by cell type. (**C**) Fold changes in cell-type abundance across conditions. Positive fold changes indicate increased abundance in Maresin 1-treated samples. *p<0.05, **p<0.01 by two-sided z test for proportions. N = 21,679 cells (10,462 cells for Maresin treatment and 11,217 cells for vehicle treatment). (**D**) Differential gene expression among each cell type across treatments normalized to the vehicle control. Gray region indicates adjusted p-value<0.05. z-scores and p-values were calculated for each gene using MAST. (**E**) Volcano plot showing differentially expressed genes among macrophages from vehicle vs Maresin 1-treated tissues. Fold changes and adjusted p-values were calculated using Wilcoxon sum-rank test. Fold change values > 2 (Log2 FC> 0.0585) and p-adjusted values < 0.05 were

*Figure 5 continued on next page*

Figure 5 continued

considered significant. (F) Volcano plot showing differentially expressed genes among MuSCs from vehicle vs Maresin 1treated tissues. Fold changes and adjusted p-values were calculated using Wilcoxon sum-rank test. Fold change values > 2 (Log2 FC > 0.0585) and p-adjusted values < 0.05 were considered significant and are plotted in yellow.

The online version of this article includes the following figure supplement(s) for figure 5:

**Figure supplement 1.** Quality control metrics for single-cell sequencing after volumetric muscle loss (VML) injury and treatment with Maresin 1.

et al., 2021a) over a time course to understand signaling factors that contribute to the observed changes in fibrosis. We detected an imbalance of pro-inflammatory eicosanoids to pro-resolving lipid mediators in degenerative VML injuries, which correlates with our previous observations of sustained inflammation and persistent neutrophil (*Larouche et al., 2022*) and macrophage (*Aguilar et al., 2018*) presence following degenerative VML. To determine whether restoring this balance of pro-inflammatory to pro-resolving lipids impacts regenerative outcomes, we administered a synthetic pro-resolving mediator Maresin 1 to mice following a degenerative VML injury. Administration of Maresin 1 attenuated fibrosis, reduced neutrophil and macrophage abundance, increased MuSC activation, and partially restored muscle force compared to vehicle-treated controls. Summing these results extends our understanding of muscle fibrosis and how targeting extracellular lipids can alter cell dynamics to manipulate this pathological outcome toward muscle regeneration.

The magnitude and duration of the inflammatory response after muscle injury is a critical determinant of healthy regeneration (*Tidball, 2017*). After VML, the immune response becomes dysregulated (*Larouche et al., 2018*) and contributes to fibrotic scarring. Eicosanoids are principal regulators of inflammation, and we detected increases in pro-inflammatory arachidonic acid-derived eicosanoids such as LTB$_4$, PGE$_2$, and PGF$_{2\alpha}$ as well as other eicosanoid metabolite pathways (5-HETE, 15-HETE) for degenerative VML through all timepoints measured. These results contrasted with detected pro-resolving lipid mediators (RvE3, 8-oxoRvD1, LXA$_4$, RvD6, PD1, PDX, and MaR1) that did not vary significantly between regenerative and degenerative VML injuries. Given neutrophils and inflammatory macrophages have been demonstrated to produce pro-inflammatory mediators in injured muscle (*Giannakis et al., 2019*), and we have previously shown persistence of neutrophils in degenerative VML for weeks after injury (*Larouche et al., 2022*), these results are consistent with increased and sustained inflammation. Our observations are distinct from muscle injuries with barium chloride (*Schmid et al., 2016*) or cardiotoxin injection (*Giannakis et al., 2019*), where inflammation subsides quickly after injury and pro-resolving lipid mediators increase in abundance. While we observed that pro-resolving lipid mediators are expressed in VML injuries, the level at which these factors are detected did not proportionally increase with injury severity and inflammation. These results suggest that while short bursts of inflammation and transient exposure to eicosanoids such COX-derived PGE$_2$ may be beneficial for muscle regeneration (*Ho et al., 2017*; *Bondesen et al., 2004*), longer-term exposure to these signaling factors and concomitant immune dysregulation without balance from pro-resolving lipid mediators (*Dort et al., 2019*; *Markworth et al., 2021b*) as observed in degenerative VML are detrimental to muscle regeneration.

The role of macrophage polarization toward a regenerative phenotype in guiding repair outcomes has been well established and is a critical determinant of development and progression of fibrosis versus successful regeneration (*Wang and Zhou, 2022*; *Wynn and Vannella, 2016*). Our results showing disproportional levels of pro-inflammatory to pro-resolving lipids following degenerative VML injury support that macrophages do not generate sufficient pro-resolving lipid mediators after injury (*Giannakis et al., 2019*; *Halade et al., 2018*) and are inhibited from effective transition (*Wang et al., 2014*; *Arnold et al., 2007*) toward a regenerative phenotype (*Chazaud, 2020*; *Hymel et al., 2021*). We recently demonstrated that the sustained inflammation post degenerative VML injury skews macrophage phenotype toward a fibrosis-promoting subtype (*Spp1*[+], *Trem2*[+], *C1qc*[+], *Cd63*[+]) (*Larouche et al., 2023*). This phenotype has also been observed in other fibrotic outcomes such as cirrhosis (*Ramachandran et al., 2019*) and obesity (*Jaitin et al., 2019*). Injection of Maresin 1, a 14S-dihydroxyl-containing pro-resolving lipid mediator that is synthesized from fatty acid docosahexaenoic acid (*Abdulnour et al., 2014*; *Li et al., 2020*), into degenerative VML-injured muscle reduced macrophage density and collagen deposition. Since fibrosis-promoting macrophages have recently been shown to be regulated by lipid metabolism, these results suggest that Maresin 1 treatment may restore lipid homeostasis and inhibit adoption of a macrophage pathological phenotype (*Henderson*

*et al., 2020*; *Jung et al., 2022*; *Deczkowska et al., 2020*). While a deeper understanding of how macrophage phenotype is manipulated by lipid concentration and type (*Sansbury et al., 2020*) VML remains to be elucidated, our results suggest pro-resolving lipid mediators may have an important role in shaping signaling that promotes macrophage differentiation toward the fibrosis-promoting subtype.

Maresin 1 signaling is mediated in part via engagement of the LGR6 receptor, which is expressed on numerous stem and progenitor cells, including in the skin (*Huang et al., 2021*), kidney (*van Ineveld et al., 2021*), and mammary gland (*Blaas et al., 2016*), and enhances proliferation, migration, and differentiation. In line with this, we observed low but detectable *Lgr6* expression in primary quiescent MuSCs, with strong upregulation during activation suggesting Maresin 1 contributes to regenerative actions of MuSCs after injury by increasing proliferation. In line with this, culture of MuSCs with Maresin 1 resulted in increased proliferation and cAMP signaling. Given canonical Wnt signaling is also a downstream target of LGR6 (*Otto et al., 2008*), we evaluated whether canonical Wnt activity was altered with Maresin 1. We did not find a change in canonical Wnt signaling within MuSCs with treatment of Maresin 1 in vitro, but did observe a strong decrease in canonical Wnt signaling in vivo. This result was consistent with reductions in macrophages and muscle fibrosis (*Larouche et al., 2018*; *Naito et al., 2012*) that promote canonical Wnt signaling through Complement. Our findings indicate that repletion of Maresin 1 positively impacts MuSCs directly through engagement of LGR6 and indirectly via alterations from myeloid signaling. However, the regenerative actions of MuSCs are still restrained from fusion due to elevated TGFβ1. We also speculate that alterations in canonical Wnt signaling in MuSCs from Maresin 1 treatment may be derived through differences in their ability to adhere to the matrix, given that β-catenin interacts with multiple cadherins (*Berdeaux and Stewart, 2012*; *Yin et al., 2013*). Additionally, Maresin 1 treatment may impact other cell types that promote functional recovery such as vasculature, which have been shown to precede myogenesis after VML injury (*Shcherbina et al., 2020*; *Jacobsen et al., 2023*). In support of this, our scRNA-seq datasets showed increased recovery of endothelial cells (*Chatterjee et al., 2014*). These observations support that Maresin 1 repletion can stimulate MuSC-mediated muscle regeneration after VML and may be promising materials to combine with other regenerative strategies (*Greising et al., 2019*).

In summary, recovery from severe muscle trauma resulting in VML is an unmet clinical need and open musculoskeletal injuries are responsible for a large fraction of hospital costs and disability payments (*Garg et al., 2015b*; *Corona et al., 2015*). The development of strategies to address lipid mediator imbalances may open new paradigms to further explore coupled immuno-regenerative (*Ho et al., 2017*; *Palla et al., 2021*) therapies.

## Materials and methods

### Animals

C57BL/6 wild-type male and female mice (3–4 months old) were obtained from Jackson Laboratory or from a breeding colony at the University of Michigan (UM). *Pax7*$^{CreERT2}$-*Rosa26*$^{Tcf/Lef-LSL-H2B-GFP}$ and *Pax7*$^{CreERT2/+}$-*Rosa26*$^{TdTomato/+}$ mice were obtained from a breeding colony at UM and administered five daily 100 uL intraperitoneal injections of 20 mg/mL tamoxifen in corn oil and allowed to recover for 2–5 d. Equal numbers of male and female mice were used for each experiment, and mice were randomly assigned into experimental groups. All mice were fed normal chow ad libitum and housed on a 12:12 hr light–dark cycle under UM veterinary staff supervision. All procedures were approved by the Institutional Animal Care and Use Committee (IACUC protocol number PRO00010663) and were in accordance with the US. National Institute of Health (NIH).

### Injury model

Mice were anesthetized with 5% isoflurane and maintained at 3% isoflurane. Buprenorphine analgesic was administered at 0.1 mg/kg dose via intraperitoneal injection prior to administering a VML injury. The surgical area was prepared by removing hair and sterilizing through series of 70% ethanol and betadine scrubbing. An incision of approximately 5 mm was administered to the skin to expose the TA muscle. A full-thickness VML injury was administered using a sterile biopsy punch of 1-mm or 2-mm diameter to the middle of the muscle followed by closure with sutures. Animals were monitored daily for 7–10 d before removing sutures.

## Tissue sectioning

After euthanasia, uninjured or injured TA muscles were harvested and embedded in an optimal cutting temperature compound and frozen in isopentane cooled with liquid nitrogen. Cross-sections were extracted from the frozen tissue blocks using a cryotome at the midpoint of the injury based on the point within the tissue where the defect was largest and delicately placed onto positively charged glass slides.

## Picrosirius staining and quantification

Tissue sections were first fixed in 4% paraformaldehyde (PFA) for 15 min at room temperature. Next, the tissue sections were washed two times with 1× phosphate-buffered saline (PBS) and followed by two washes with deionized (DI) water. The sections were then air-dried for 20 min and stained with Sirius red dye for 1 hr in a humidifying chamber. Sirius red dye was washed with DI water one time for 5 min followed by sequential dehydration immersions in 50, 70, 70, 90, and 100% ethanol solutions, and two 5 min incubation in xylenes at room temperature. Coverslips were mounted with Permount, and whole section images were imaged using a motorized Olympus IX83 microscope. Area of the tissue section and collagen area were quantified from the stitched images by automated thresholding in FIJI or MATLAB. Collagen percentage was calculated as (collagen area ÷ tissue section area) * 100 and graphed in GraphPad. Sample size was determined by power analysis of pilot experiments to achieve 80% power.

## Metabolipidomics

C18 columns were conditioned using 15% methanol and hexane. Elutions were performed by doing two washes using 100% methanol and dried using a gentle stream of nitrogen gas. After resuspending dried elutions in 50 µL of methanol-25 mM aqueous ammonium acetate (1:1), LC-MS/MS was performed in a prominence XR system (shimaduzu) using Luna C18 columns. LC-MS/MS data were analyzed using MetaboAnalyst 4.065.

## Processing of lipid abundance data

Raw lipid abundances were normalized and prepared for downstream analyses using the MetaboDiff package (*Mock et al., 2018*) (v0.9.5) in R (v4.2.1). Outlier samples were identified using PCA and removed, and knn imputation was repeated for the remaining samples. The data was then subjected to variance stabilization normalization (vsn) for downstream processing.

## Differential lipid abundances analysis

One-way ANOVAs (aov command in R) were performed for each lipid for the injured timepoints with the following design formula: Concentration ~ Condition, where Condition = {Injury +Time}, Injury = {1 mm, 2 mm}, and Time = {0, 3, 7, 14 dpi}. Differential lipids were identified as those with p-values <0.05 after Benjamini–Hochberg correction.

## DPGP clustering

To cluster differentially abundant lipids by similar abundance dynamics over the time course, we used the Dirichlet Process Gaussian Process mixture model (DPGP v0.1) (*McDowell et al., 2018a*). Normalized imputed abundances were averaged within each condition and fold changes were calculated between injuries (2 mm over 1 mm) at each timepoint. Fold changes for each lipid across time points were normalized as z-scores, then clustered with DPGP using default parameters with the following command:

```
DP_GP_cluster.py -i [fold change z-scores] -o [output file prefix]
```

## Immunohistochemistry stain and quantification

Tissue sections were thawed and air-dried at room temperature for 5 min followed by fixation in 100% acetone cooled to –20°C for 10 min or 4% PFA in PBS at room temperature for 15 min. Tissues were rehydrated with 1× PBS for 5 min and blocked in 10% goat serum diluted in 1× PBS for 1 hr or MOM blocking reagent (Vector Labs) for 30 min. Primary antibodies (CD68 Bio-Rad, MYH3 DHSB, GFP Abcam, RFP Rockland) were incubated overnight at 4°C in a humidifying chamber. Secondary

antibodies (Thermo Fisher) were incubated for 1 hr at room temperature in a humidifying chamber. Coverslips were mounted using Prolong Diamond Antifade. Whole-section images were acquired using a Nikon A1 confocal microscope. The CD68+ area was quantified as a percentage of the full-section area using MATLAB. Myofiber regions were determined using Cellpose (*Stringer et al., 2021*) and LabelsToROIs (*Waisman et al., 2021*), followed by quantification of MYH3+ fibers and MYH3+ fiber measurements using MATLAB. Three sections per tissue were imaged, quantified, and averaged and graphed in GraphPad prism. To quantify TdTomato+ myofibers, Cellpose (*Stringer et al., 2021*) and LabelsToROIs (*Waisman et al., 2021*) were used to determine myofiber regions followed by quantification of TdTomato+ fibers using FIJI. TdTomato+ fiber measurements were processed using R and graphed in GraphPad prism. Sample size was determined by power analysis of pilot experiments to achieve 80% power.

## Flow cytometry

Mouse TAs were extracted, separately weighed using sterile surgical tools, and minced using surgical scissors. Minced tissues were added to tubes containing collagenase type II (0.2%) and dispase II (2.5 U/mL) in 10 mL of DMEM, then placed on rocker in a 37°C incubator for 1 hr and mixed by pipette every 30 min. The enzymes were then inactivated by addition of 20% heat-inactivated fetal bovine serum (HI-FBS) in Ham's F10 media, and the solution was passed through a 70-um cell strainers, centrifuged, washed, and counted. The single-cell suspension was then pelleted and resuspended in staining buffer (PBS with 2% BSA, 2 mM EDTA and 0.01% sodium azide), and plated in a 96-well-round bottom plate. Cells were centrifuged at $350 \times g$ for 2.5 min and resuspended in a primary antibody cocktail including CD45-APC (BioLegend), F4/80-AF488 (BioLegend), and Ly6G-APCCy7 (BioLegend) for 30 min on ice. Cells were then centrifuged, washed with staining buffer, then resuspended in staining buffer containing propidium iodide for 1 min at room temperature in the dark, centrifuged, and resuspended in staining buffer for flow cytometry analysis. Prior to acquisition, cells were filtered through 40-mm cell strainers. Single-color controls were made using UltraComp eBeads (Thermo Fisher) compensation beads stained according to the manufacturer's protocol. Samples were acquired within 30 min on a Bio-Rad Ze5 cytometer, and the data was processed using FlowJo (version 10) with manual compensation. Sample size was determined by power analysis of pilot experiments to achieve 80% power.

## MuSC enrichment

Mouse TA muscles were extracted and separately weighed using sterile surgical tools and placed into separate Petri dishes containing ice-cold PBS. Using surgical scissors, muscle tissues were minced and collagenase type II (0.2%) and dispase II (2.5 U/mL) were added to 10 mL of DMEM per quadricep. Samples were placed on rocker in a 37°C incubator for 1.5 hr and mixed by pipette every 30 min. The enzymes were then inactivated by addition of 20% HI-FBS in Ham's F10 media. The solution was passed through a 70-um cell strainers, centrifuged, and washed. Negative MuSC enrichment was performed using the Miltenyi Satellite Cell Isolation Kit for mouse according to the manufacturer's protocols or by FACS sorting for CD45-Ter119-CD31-CD11b-Sca1-B1int+ CXCR4+ cells as previously described (*Shcherbina et al., 2020*).

## Quantitative real-time PCR (qRT-PCR)

Cells were lysed directly in the plate using Buffer RLT (QIAGEN) with 10 uL/mL 2-mercaptoethanol following a quick PBS rinse. Cell lysates were thawed at room temperature for 30 min, then RNA was extracted using the QIAGEN RNeasy Mini Kit according to the manufacturer's instructions. RNA purity and concentration were measured using a NanoDrop and Qubit RNA HS Assay. Within 1 wk, cDNAs were synthesized using the SuperScript III cDNA Synthesis Kit (Thermo Fisher) according to the manufacturer's protocol. DNA quality and concentration were determined using a NanoDrop. Then, 80–100 ug cDNA template was plated in triplicate along with SYBR Green PCR MasterMix and 500 nM PCR primer, then cycled 40 times starting at 95°C for 10 s followed by 60°C for 30 s on a CFX96 Real-Time thermocycler. Gene expression was quantified using the ΔΔCt method.

## In vivo Maresin 1 treatment

Maresin 1 (Cayman Chemicals #1268720-28-0) was aliquoted (500 ng per vial) in amber glass vials (Thermo Fisher Scientific, C4010-88AW), purged with nitrogen gas, and stored at –80°C. On the day

of use, ethanol was evaporated using a stream of nitrogen gas and resuspended in 100 μL of sterile saline solution + 0.1% ethanol. Maresin 1 solution was protected from light and kept on ice until intramuscular administration. Mice legs were randomized to receive 100 ng of Maresin 1 (20 μl) or vehicle (sterile saline + 0.1% ethanol) every 2 d after injury.

## In vitro Maresin 1 treatment

Single-cell suspensions were prepared as described above in sterile conditions, followed by MACS isolation of MuSCs using the Miltenyi Satellite Cell Isolation Kit for mouse according to the manufacturer's protocols. MACS-enriched MuSCs were plated on Matrigel-coated tissue culture dishes in myoblast media (Ham's F10 with 20% FBS, antibiotics, and fibroblast growth factor basic) and allowed to expand for up to three passages. Myoblasts were then passaged and plated in a 48-well plate with 15,000 cells seeded per well. After 24 hr, media was replaced with fresh myoblast media containing Maresin 1 (10 nM) and EdU (10 uM). EdU staining was performed 24 hr later using the EdU ClickIt Reaction Kit (Thermo Fisher) according to the manufacturer's instructions. Images were analyzed using MATLAB to quantify the percentage of cells positive for EdU incorporation.

## In vitro Lgr6 knockdown

MuSCs were MACS-enriched and cultured as described above. Following expansion in myoblast media (F10 with 10% HI FBS, bFGF, and antibiotics), cells were seeded into 12-well plates at a density of 50,000 cells per well. Lgr6 knockdown was performed using RNAiMAX (Thermo Fisher) and Lgr6 Dsi-RNA (IDT) according to the manufacturer's protocol in myoblast media without antibiotics. After 72 hr, cells were either lysed in the plate for RT-qPCR validation of knockdown efficacy, or media was replaced with myoblast media containing EdU for proliferation analysis, or with differentiation media (DMEM containing 5% horse serum and antibiotics). EdU staining was performed after 24 hr in EdU according to the manufacturer's protocols. Wnt signaling activation was assessed using cells isolated from P7Wnt mice. After 72 hr in differentiation media, cells were washed, stained with Hoechst 33342, and imaged on a Zeiss epifluorescent microscope using a ×20 objective. GFP mean fluorescent intensity among GFP+ DAPI+ cells was quantified using MATLAB.

## In vitro P7Wnt mouse validation

MuSCs were isolated from P7Wnt mice via MACS and expanded in myoblast media. Cells were seeded into a 48-well plate at a density of 15,000 cells per well. After allowing the cells 24 hr to adhere to the well plate, media was replaced with fresh myoblast media containing 50 ng/mL recombinant Wnt3a protein (R&D Systems #1324-WN-002). MATLAB was used to quantify the percentage of GFP+ cells.

## Single-cell RNA sequencing

### Sample preparation and sequencing

Male and female mice received bilateral 2-mm TA VML defects, which were treated with 20 uL intramuscular injections of Maresin 1 (100 ng in 0.1% EtOH in saline) or vehicle (0.1% EtOH in saline) at 1, 3, and 5 dpi. Animals were euthanized as described above at 7-dpi. TA muscles were pooled from three mice according to treatment, then digested into single-cell suspensions as described above. Labeling with cell multiplexing oligos (CMOs) (10x Genomics) was performed according to the manufacturer's instructions (Demonstrated Protocol CG000391 Rev B, Protocol 3). Each treatment condition (vehicle and Maresin 1) was labeled with a separate CMO tag. Then equal cell numbers from each sample were pooled, stained with 7-AAD, and FACS-sorted to remove dead cells and debris. Post FACS, 8000 cells were loaded into the 10x Genomics chromium single-cell controller, and single cells were captured into nanoliter-scale gel bead-in-emulsions (GEMs). cDNAs were prepared using the single-cell 3' protocol as per the manufacturer's instructions and sequenced on a NovaSeq 6000 (Illumina) with 26 bases for read1 and 98 × 8 bases for read2.

### Data processing and analysis

10x CellRanger v7.0.0 software's mkfastq and multi command were run with default parameters except expect-cells = 8000. HD5 files were imported into R v.4.2.1 (https://www.r-project.org/) using the Seurat (*Otto et al., 2008*) v4.2 package, and genes expressed in less than 200 cells or cells expressing less than three genes were removed. Seurat objects were then merged. Normalization

was performed and variable features were identified on each dataset separately prior to integration using Seurat v4. Linear dimensional reduction was performed using RunPCA, followed by FindNeighbors(dims = 1:30) and RunUMAP(32) (dims = 1:30). Clustering was performed using the Louvain unsupervised clustering algorithm at a resolution = 0.1. Cluster marker genes were determined using Seurat's FindAllMarkers function (only.pos = T, logfc.threshold = 1) to annotate cell types. Differential gene expression across treatments within cell types was performed using MAST (*Jacobsen et al., 2023*). Seurat, dittoSeq (*Chatterjee et al., 2014*), EnhancedVolcano, and ggplot2 were used for data visualization.

## In situ functional testing

These procedures are modified from (*Dellorusso et al., 2001*). Briefly, mice were anesthetized with intraperitoneal injections of tribromoethanol (250 mg/kg) and supplemental injections given to maintain an adequate level of anesthesia during the procedure. Hindlimb fur was removed with animal clippers. The TA muscle was exposed by removing the overlying skin and outer fasciae. The distal TA tendon was isolated, and the distal half of the TA was freed from adjacent muscles by carefully cutting fasciae without damaging muscle fibers. A 4-0 silk suture was tied around the distal tendon, and the tendon was severed. The animal was then placed on a temperature-controlled platform warmed to maintain body temperature at 37°C. A 25-gauge needle was driven through the knee and immobilized to prevent the knee from moving. The tendon was tied securely to the lever arm of a servomotor via the suture ends (6650LR, Cambridge Technology). A continual drip of saline warmed to 37°C was administered to the TA muscle to maintain temperature. The TA muscle was initially stimulated with 0.2 ms pulses via the peroneal nerve using platinum electrodes. Stimulation voltage and muscle length were adjusted for maximum isometric twitch force (Pt). While held at optimal muscle length (Lo), the muscle was stimulated at increasing frequencies until a maximum force (Po) was reached, typically at 200 Hz, with a 1 min rest period between each tetanic contraction. Subsequently, the same procedure was repeated, but rather than activating the muscle via the peroneal nerve, a cuff electrode was placed around the muscle for stimulation. Muscle length was measured with calipers, based on well-defined anatomical landmarks near the knee and the ankle. Optimum fiber length (Lf) was determined by multiplying Lo by the TA Lf/Lo ratio of 0.6. After the evaluation of isometric force, the TA muscle was removed from the mouse. The tendon and suture were trimmed from the muscle, and the muscle was weighed. Total muscle fiber CSA of TA muscles was calculated by dividing muscle mass by the product of Lf and 1.06 mg/mm$^3$, the density of mammalian skeletal muscle (*Mendez and Keys, 1960*). Specific Po was calculated by dividing Po by CSA. Sample size was determined by power analysis of pilot experiments to achieve 80% power.

## TGFβ1 ELISA

Muscles were extracted at 7 dpi as described above and flash frozen in liquid nitrogen, then stored at –80°C. Tissues were thawed in ice-cold PBS, weighed, minced, and homogenized with 30 passes of a Dounce homogenizer in 500 uL of RIPA buffer (Thermo Fisher) with a protease inhibitor cocktail (Thermo Fisher). Total protein was quantified using a Pierce BCA Assay kit (Thermo Fisher) according to the manufacturer's instructions. Active TGFβ1 was quantified using the mouse TGF beta 1 DuoSet ELISA kit (R&D Systems) according to the manufacturer's instructions. Absorbances were measured on a Synergy Neo microplate reader.

## cAMP assay

Hindlimb muscles were extracted, weighed, and minced using sterile surgical tools. Minced tissue was digested as previously described during the flow cytometry preparation. MuSCs were isolated via MACS (Miltenyi Biotec #130-104-268) and expanded in myoblast media. Primary myoblasts were then cultured in Matrigel-coated 96-well plates (Thermo Fisher #165306), seeding 2500 cells per well (three wells per biological replicate). Cells were incubated with Maresin 1 (10 nM) or vehicle for 24 hr. The cAMP assay (Promega #V1501) was then run, and luminescence levels were measured on a Synergy Neo microplate reader. cAMP levels were calculated based on a nonlinear fit (agonist vs response) of the standard curve.

## Statistics

Experiments were repeated at least twice, apart from scRNA-seq. Bar graphs show mean ± standard error from biological replicates unless otherwise stated. Statistical analysis was performed in GraphPad and/or R using two-sample Student's *t*-test assuming normal distribution and equal variances, one-way ANOVA, or paired-*t*-test, as specified in the figure captions. All statistical tests performed were two-sided. Outliers were determined using the IQR method and removed from further analysis. p-Values<0.05 were considered statistically significant.

## Acknowledgements

The authors thank the University of Michigan DNA Sequencing Core for assistance with sequencing and Jeremy Nathans for providing the *Rosa26-Tcf/Lef-LSL-H2B-GFP* strain. The research reported in this publication was partially supported by the National Institute of Arthritis and Musculoskeletal and Skin Diseases of the National Institutes of Health under Award Number P30 AR069620 (CAA and SVB), Genentech Research Award (CAA), the 3M Foundation (CAA), American Federation for Aging Research Grant for Junior Faculty (CAA), the Department of Defense and Congressionally Directed Medical Research Program W81XWH2010336 and W81XWH2110491 (CAA), a National Science Foundation CAREER award (2045977), Defense Advanced Research Projects Agency (DARPA) 'BETR' award D20AC0002 (CAA) awarded by the US Department of the Interior (DOI), Interior Business Center, Hevolution HF-AGE award (CAA), the Howard Hughes Medical Institute through the James H Gilliam Fellowships for Advanced Study program (GT15755 to JACM), and the National Science Foundation Graduate Research Fellowship Program under Grant Number DGE 1256260 (JAL). The content is solely the responsibility of the authors and does not necessarily represent the official views of the National Institutes of Health or National Science Foundation, the position or the policy of the Government, and no official endorsement should be inferred.

## Additional information

### Funding

| Funder | Grant reference number | Author |
| --- | --- | --- |
| National Institute of Arthritis and Musculoskeletal and Skin Diseases | P30 AR069620 | Carlos A Aguilar |
| Congressionally Directed Medical Research Programs | W81XWH2010336 | Carlos A Aguilar |
| Congressionally Directed Medical Research Programs | W81XWH2110491 | Carlos A Aguilar |
| 3M Foundation | | Carlos A Aguilar |
| American Federation for Aging Research | | Carlos A Aguilar |
| National Science Foundation | 2045977 | Carlos A Aguilar |
| Defense Advanced Research Projects Agency | D20AC0002 | Carlos A Aguilar |
| Hevolution Foundation | | Carlos A Aguilar |

The funders had no role in study design, data collection and interpretation, or the decision to submit the work for publication.

## Author contributions
Jesus A Castor-Macias, Jacqueline A Larouche, Conceptualization, Data curation, Formal analysis, Investigation, Methodology, Writing - original draft, Writing - review and editing; Emily C Wallace, Bonnie D Spence, Alec Eames, Benjamin A Yang, Carol A Davis, Susan V Brooks, Krishna Rao Maddipati, Investigation, Methodology; Pamela Duran, Paula M Fraczek, Methodology; James F Markworth, Conceptualization, Data curation, Formal analysis, Supervision, Investigation, Methodology, Writing - original draft, Project administration, Writing - review and editing; Carlos A Aguilar, Conceptualization, Formal analysis, Supervision, Investigation, Methodology, Writing - original draft, Project administration, Writing - review and editing

## Author ORCIDs
Jesus A Castor-Macias http://orcid.org/0000-0002-2501-1035
Jacqueline A Larouche http://orcid.org/0000-0001-9380-3547
James F Markworth http://orcid.org/0000-0002-5348-1464
Carlos A Aguilar http://orcid.org/0000-0003-3830-0634

## Ethics
All mice were fed normal chow ad libitum and housed on a 12:12 hour light-dark cycle under UM veterinary staff supervision. Allprocedures were approved by the Institutional Animal Care and Use Committee (IACUC) andwere in accordance with the U.S. National Institute of Health (NIH).

## Decision letter and Author response
Decision letter https://doi.org/10.7554/eLife.86437.sa1
Author response https://doi.org/10.7554/eLife.86437.sa2

## Additional files

### Supplementary files
• MDAR checklist

• Supplementary file 1. Metabolite concentration (pmol/g) for different pathways in uninjured, 1 mm VML, and 2 mm (TA) tissue homogenates.

### Data availability
Sequencing data have been deposited in GEO under accession codes GSE215808.

The following dataset was generated:

| Author(s) | Year | Dataset title | Dataset URL | Database and Identifier |
|---|---|---|---|---|
| Castor-Macias JA, Larouche JA, Wallace EC, Spence BD, Eames A, Yang BA, Davis C, Brooks SV, Maddipati KR, Markworth JF, Aguilar CA | 2023 | Maresin 1 repletion improves muscle regeneration after volumetric muscle loss | http://www.ncbi.nlm.nih.gov/geo/query/acc.cgi?acc=GSE215808 | NCBI Gene Expression Omnibus, GSE215808 |

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

# Appendix 1

## Appendix 1—key resources table

| Reagent type (species) or resource | Designation | Source or reference | Identifiers | Additional information |
|---|---|---|---|---|
| Biological sample (*Mus musculus*) | C57BL/6J wild-type mice | Jackson Labs | Jackson Stock 000664 | Female (2–3 mo) |
| Biological sample (*M. musculus*) | *Pax7Cre<sup>ERT2/+</sup>;Rosa26<sup>Tcf/Lef-LSL-H2B-GFP</sup>* mice | University of Michigan | Jackson stock 017763 crossed with Jackson stock 032577 | Female (2–3 mo) |
| Biological sample (*M. musculus*) | *Pax7<sup>CreERT2</sup> – Rosa26<sup>-TdTomato</sup>* | University of Michigan | Jackson stock 017763 crossed with Jackson stock 007914 | Female (2–3 mo) |
| Antibody | AF488 anti-mouse monoclonal F4/80, clone: BM8, isotype: Rat IgG2a, κ | BioLegend | 123119; RRID:AB_893491 | FC (1:200) |
| Antibody | APC-Cy7 anti-mouse monoclonal Ly-6G, clone: 1A8, isotype: Rat IgG2a, κ | BioLegend | 127624; RRID:AB_10640819 | FC (1:400) |
| Antibody | APC anti-mouse monoclonal Ly-6A/E (Sca-1), clone: D7, isotype: Rat IgG2a, κ | BioLegend | 108112; RRID:AB_313349 | FC (1:400) |
| Antibody | APC anti-mouse monoclonal CD45, clone: 30-F11, isotype: Rat IgG2b, κ | BioLegend | 103112; RRID:AB_312977 | FC (1:400) |
| Antibody | APC anti-mouse monoclonal TER-119, clone: TER-119, isotype: Rat IgG2b, κ | BioLegend | 116212; RRID:AB_313713 | FC (1:400) |
| Antibody | APC anti-mouse monoclonal CD31, clone: 390, Isotype: Rat IgG2a, κ | BioLegend | 102410; RRID:AB_312905 | FC (1:400) |
| Antibody | APC anti-mouse/human monoclonal CD11b, clone: M1/70. Isotype: Rat IgG2b, κ | BioLegend | 101212; RRID:AB_312795 | FC (1:400) |
| Antibody | PE anti-mouse/rat monoclonal CD29 (B1 int), clone: HMβ1-1, isotype: Armenian Hamster IgG | BioLegend | 102208; RRID:AB_312885 | FC (1:200) |
| Antibody | Biotin Rat AntiMouse monoclonal CD184, clone: 2B11/CXCR4 (RUO), isotype: Rat IgG2b, κ | BD Biosciences | 551968; RRID:AB_394307 | FC (1:200) |
| Antibody | Anti-mouse monoclonal CD68, clone: FA-11, isotype: IgG2a | Bio-Rad | MCA1957 | IF (1:50) |
| Antibody | Mouse monoclonal Anti-human MYH3 | DHSB | F1.652 | IF (1:20) |
| Antibody | Chicken polyclonal Anti-mouse GFP | Abcam | ab13970; RRID:AB300798 | IF (1:1000) |
| Antibody | Rabbit anti-mouse laminin 1+2, Isotype: Polyclonal IgG | Abcam | ab7463; RRID:AB_305933 | IF (1:500) |
| Antibody | Rabbit polyclonal Anti-RFP | Rockland | 600-401-379; RRID:AB_2209751 | IF (1:50) |
| Antibody | Rabbit polyclonal Anti-Beta catenin polyclonal (CAT-15) | Thermo Fisher | 71-2700; RRID:AB_2533982 | IF (1:100) |
| Antibody | Goat polyclonal Anti-rabbit IgG (H+L), Alexa Fluor 647 conjugate | Thermo Fisher | A27040; RRID:AB_2536101 | IF (1:500) |
| Antibody | Goat polyclonal Anti-rat (H+L), Alexa Fluor 647 conjugate | Thermo Fisher | A21247; RRID:AB_141778 | IF (1:500) |
| Antibody | Goat polyclonal Anti-mouse (H+L), Alexa Fluor 488 conjugate | Thermo Fisher | A28175; RRID:AB_2536161 | IF (1:500) |
| Antibody | Goat polyclonal Anti-chicken IgY (H+L), Alexa Fluor 488 conjugate | Thermo Fisher | A32931TR; RRID:AB_2866499 | IF (1:500) |

*Appendix 1 Continued on next page*

*Appendix 1 Continued*

| Reagent type (species) or resource | Designation | Source or reference | Identifiers | Additional information |
|---|---|---|---|---|
| Antibody | Goat polyclonal Anti-rabbit (H+L), Alexa Fluor 555 conjugate | Thermo Fisher | A21247; RRID:AB_141778 | IF (1:500) |
| Sequence-based reagent | PrimeTime Mouse GAPDH Primer | Integrated DNA Technologies | Mm.PT.39a1 | |
| Sequence-based reagent | PrimeTime Mouse Lgr6 Primer | Integrated DNA Technologies | Mm.PT.58.9348010 | |
| Sequence-based reagent | Mouse Lgr6 DsiRNA #1 | Integrated DNA Technologies | mm.Ri.Lgr6.13.2 | |
| Sequence-based reagent | Mouse Lgr6 DsiRNA #2 | Integrated DNA Technologies | mm.Ri.Lgr6.13.1 | |
| Sequence-based reagent | Cell Multiplexing Oligos | 10x Genomics | 1000261 | |
| Peptide, recombinant protein | Bovine serum albumin | Fisher Scientific | BP9703-100 | |
| Peptide, recombinant protein | Fibroblast growth factor basic | Gibco-Invitrogen | PHG0263 | |
| Peptide, recombinant protein | 0.25% Trypsin EDTA | Gibco-Invitrogen | 25200072 | |
| Peptide, recombinant protein | HALT Protease Inhibitor Cocktail | Thermo Fisher | 87786 | |
| Chemical compound, drug | Dispase II (activity ≥ 0.5 units/mg solid) | Sigma | D4693-1G | Digestion of tissue to extract MuSCs |
| Chemical compound, drug | Collagenase type II (654 U/mg, non-specific proteolytic activity 487 U/mg) | Life Technologies | 17101015 | Digestion of tissue to extract MuSCs |
| Chemical compound, drug | DMEM, high glucose, pyruvate | Life Technologies | 11995065 | |
| Chemical compound, drug | Ham's F-10 Nutrient Mix | Life Technologies | 11550043 | |
| Chemical compound, drug | Tissue Plus O.C.T Compound | Fisher Scientific | 23-730-571 | |
| Chemical compound, drug | Magnesium sulfate heptahydrate | Sigma-Aldrich | 63138-250G | |
| Chemical compound, drug | Sodium bicarbonate | Sigma-Aldrich | S5761 | |
| Chemical compound, drug | SafeClear II | Fisher Scientific | 23-044192 | |
| Chemical compound, drug | Picric acid | Sigma-Aldrich | P6744-1GA | |
| Chemical compound, drug | Glacial acetic acid | Sigma-Aldrich | BP2401-500 | |
| Chemical compound, drug | Xylenes | Sigma-Aldrich | 534056-4L | |
| Chemical compound, drug | Permount | Fisher Scientific | SP15-100 | |
| Chemical compound, drug | 0.5 M EDTA | Invitrogen | 15575-038 | |
| Chemical compound, drug | Sodium azide | Sigma-Aldrich | 71289 | |
| Chemical compound, drug | 4% paraformaldehyde in PBS | Thermo Fisher | J19943-K2 | |

*Appendix 1 Continued on next page*

*Appendix 1 Continued*

| Reagent type (species) or resource | Designation | Source or reference | Identifiers | Additional information |
|---|---|---|---|---|
| Chemical compound, drug | Penicillin streptomycin | Gibco-Invitrogen | 15640055 | |
| Chemical compound, drug | Maresin 1 | Cayman Chemicals | 10878 | |
| Chemical compound, drug | Tween-20 | Sigma-Aldrich | P1379 | |
| Chemical compound, drug | TritonX-100 | Sigma-Aldrich | T8787 | |
| Chemical compound, drug | Tamoxifen | Sigma-Aldrich | T2859-1G | |
| Chemical compound, drug | Corn Oil | Sigma-Aldrich | C8267-2.5L | |
| Commercial assay or kit | Satellite Cell Isolation Kit, mouse | Miltenyi | 130-104-268 | |
| Commercial assay or kit | TGF beta 1 DuoSet ELISA Kit | R&D Systems | DY1679 | |
| Commercial assay or kit | Pierce BCA Assay Kit | Thermo Fisher | 23227 | |
| Commercial assay or kit | EdU ClickIt Reaction Kit | Thermo Fisher | C10340 | |
| Commercial assay or kit | SuperScript III First-Strand Synthesis Kit | Thermo Fisher | 18080051 | |
| Commercial assay or kit | QIAGEN RNeasy Mini Kit | QIAGEN | 74104 | |
| Commercial assay or kit | Qubit RNA HS Assay | Thermo Fisher | Q32852 | |
| Commercial assay or kit | Single cell 3' Library & Gel Bead Kit | 10x Genomics | 120267 | |
| Commercial assay or kit | cAMP-Glo Assay | Promega | V1501 | |
| Software, algorithm | CellRanger v7.0.0 | 10x Genomics | https://support.10xgenomics.com/single-cell-gene-expression/software/downloads | |
| Software, algorithm | R v4.2.1 | The R Foundation for Statistical Computing | https://www.r-project.org/; RRID:SCR_001905 | |
| Software, algorithm | Python | | https://www.python.org/ | |
| Software, algorithm | MetaboDiff v0.9.5 | *Mock et al., 2018*; *Mock, 2020* | https://github.com/andreasmock/MetaboDiff | |
| Software, algorithm | DPGP v0.1 | *McDowell et al., 2018a*; *McDowell et al., 2018b* | https://github.com/PrincetonUniversity/DP_GP_cluster | |
| Software, algorithm | MATLAB_R2020a | MathWorks | https://www.mathworks.com/products/matlab.html | |
| Software, algorithm | Seurat v4.2.1 | *Stuart et al., 2019* | https://satijalab.org/seurat/; RRID:SCR_007322 | |
| Software, algorithm | ggplot2 v3.2.1 | *Wickham et al., 2016* | https://ggplot2.tidyverse.org; RRID:SCR_014601 | |
| Software, algorithm | FlowJo v10 | | https://www.flowjo.com; RRID:SCR_008520 | |
| Software, algorithm | ImageJ v2.1.0 | | https://imagej.net/ImageJ; RRID:SCR_002285 | |
| Software, algorithm | Cellpose | *Stringer et al., 2021* | RRID:SCR_021716 | |
| Software, algorithm | LabelsToROIs | *Waisman et al., 2021* | https://labelstorois.github.io/ | |

*Appendix 1 Continued on next page*

*Appendix 1 Continued*

| Reagent type (species) or resource | Designation | Source or reference | Identifiers | Additional information |
|---|---|---|---|---|
| Software, algorithm | Bioinformatics analysis code | This paper | https://github.com/AguilarLab/Maresin1 (copy archived at *Castor-Macias et al., 2023*) | Algorithm used to generate *Figure 5*, *Figure 5—figure supplement 1*. Refer to data processing and analysis under single-cell RNA sequencing |
| Other | Streptavidin PE-Cyanine7 | Thermo Fisher | 25-4317-82; RRID:AB_10116480 | FC (1:100) |
| Other | scRNA-seq datasets | This paper | GSE215808 | Refer to data processing and analysis under single-cell RNA sequencing. |
| Other | LS Columns | Miltenyi | 130-042-401 | MACS isolation |
| Other | UltraComp eBeads | Fisher Scientific | 01-2222-42 | Single color controls for Flow Cytometry |
| Other | RIPA Buffer | Thermo Fisher | 89900 | Tissue homogenization reagent in ELISA assays |
| Other | Lipofectamine RNAiMAX | Invitrogen | 13-778-030 | Transfection reagent |
| Other | Hoechst 33342 | Thermo Fisher | H3570 | IF (1:1000) nuclear stain |
| Other | SYBR Green PCR MasterMix | Thermo Fisher | 4309155 | PCR reagent |
| Other | Matrigel | BD Biosciences | 356234 | Cell culture coating reagent |
| Other | Horse Serum | Gibco-Invitrogen | 26050088 | Cell culture |
| Other | Prolong Diamond | Thermo Fisher | P36965 | Mounting media |
| Other | Direct Red 80 | Fisher Scientific | AAB2169306 | Picrosirius red stain |
| Other | Hematoxylin | Ricca Chemical Company | 3530-16 | H&E stain |
| Other | Eosin | EMD-Millipore | 588X-75 | H&E stain |
| Other | Fetal bovine serum | Life Technologies | 10437028 | Tissue digestion reagent |
| Other | Normal goat serum | Abcam | Ab7481; RRID:AB_2716553 | IHC stain |
| Other | Propidium iodide – 1.0 mg/mL solution in Water | Life Technologies | P3566 | FC (1:400) |
| Other | 7-AAD | BioLegend | 420403 | FC (1:200) |
| Other | Mouse on Mouse blocking reagent | Vector Labs | MKB-2213 | IHC stain |

