## [Editor Report]

This is a compelling study on pro-inflammatory and anti-inflammatory lipids in relation to skeletal muscle injury. It convincingly identifies pro-inflammatory lipids during recovery predisposing to fibrosis, and maresin 1 as an anti-inflammatory lipid reducing fibrosis, improved muscle regeneration, partially restoring contractile function, of fundamental potential clinical applicagtion.

---

## [Decision Letter]

**Decision letter after peer review:**

Thank you for submitting your article "Maresin 1 Repletion Improves Muscle Regeneration After Volumetric Muscle Loss" for consideration by *eLife*. Your article has been reviewed by 2 peer reviewers, and the evaluation has been overseen by a Reviewing Editor and Mone Zaidi as the Senior Editor. The following individual involved in the review of your submission has agreed to reveal their identity: Steven Segal (Reviewer #1).

Both reviewers have responded positively to the novelty and value of your work and its new insights into the roles of lipid mediators in VML injuries. They consider your new data of good quality. They recognize its potential to guide the development of therapeutic interventions promoting tissue repair and recovery. The reviewers have discussed their reviews with one another, and the Reviewing Editor has drafted this to help you prepare a revised submission.

Essential revisions:

Reviewer (1) has the following general comments:

Certain aspects of the data are overinterpreted; while some measures appear to have an adequate sample size to make sound conclusions, other measures are likely to lack sufficient statistical power given their variability. Presentation of the results would be strengthened by adhering to consistent terminology and labeling of figures throughout; specific examples are identified in recommendations to the authors. Several of the images used to illustrate differences between treatments are unconvincing because differences are not readily.

Reviewer (2) raises the following specific issues that require addressing:

1) The authors use 1mm vs 2mm punch biopsies to create "regenerative" vs "degenerative" injuries in the mouse tibialis anterior muscle. In a key reference (25), the threshold for successful regeneration following a biopsy of the mouse quadriceps muscle was defined as 3 mm; below this value (e.g., 2 mm) regeneration was found to be successful. Does the size of the injury for effective regeneration vary with the size of the muscle?

2) The authors introduce "dpi" to denote days post-injury. However, this acronym is not used consistently either in the text, figures, or figure captions (see comment 9 below).

3) The difference between legs in the size of the defect is described in the caption of Figure 2 but the difference is illustrated in Figure 1. Recommend revising the caption to Figure 1 accordingly. The notation used at the top of panel B is inconsistent.

4) It is not clear when (dpi) some of the data in the figures were obtained. A good example is panels C, E, and F in Figure 3, where time points are given for other panels but not these. Recommend perusal and revision throughout the text and figures (captions) for consistency. It is also not clear why key variables were not measured at respective time points (e.g., collagen deposition at 28 dpi when force was evaluated).

5) It is not clear which colors represent EdU incorporation or Wnt expression in Figure 4; for those not familiar with respective techniques, it would be helpful to identify these details. A specific concern is that the images in panels B and D are not convincing with respect to illustrating clear differences between 0 and 10 nM Maresin 1 treatment in B, or between Control and Knockdown in D.

6) An additional concern exemplified by the data in Panels C and E of Figure 4 is the low sample size (n=3), particularly given the variability in the data. There is no mention of tests for statistical power in the manuscript. It is recommended that such tests be applied for each of the variables upon which conclusions are based and the sample size adjusted accordingly to strengthen the conclusions presented to the reader.

7) In Figure 5, the UMAP plots for Vehicle and Maresin 1 are indistinguishable, making it difficult for the reader to identify the differences indicated in the text. Further, the text describing these data on p.5 indicates that respective conditions represent a pool of 3 tissues from 3 mice. It is not clear how statistics can be performed by comparing one pooled sample vs another pooled sample. It would also be helpful to clarify the timepoint (dpi) for these data in the figure caption.

8) It is remarkable that what is presented as a statistically significant difference for endothelial cells in Figure 5B is not addressed in the text. A recent study in (10.1186/s13395-023-00313-3) points to angiogenesis (i.e., endothelial cell proliferation) as an integral component of regeneration following a punch biopsy injury.

9) Panel A in Supplemental Figure 2 is confusing. There are more colors for respective clusters than are indicated on the Y-axis. Per comment 2), there are two different formats used here to identify time points, neither of which is dpi. Recommend revising all figures, captions, and text to use dpi for consistency. It may be helpful to define uninjured as 0 dpi.

10) Text in the Discussion associated with "at this timepoint" (p.6, paragraph 1, line 18) is confusing. Clarify intent.

11) Text in the Discussion associated with "did not vary significantly with larger injury sizes" (p.6, paragraph 2, line 8) is confusing. Clarify intent.

12) In Supplemental Figure 1, Isometric Force is presented as "Tetanus" throughout the range of stimulation frequency in panels A and B. Clarify what is going on at lower frequencies and which values are used in panels C, D, and E. It would also be helpful to clarify why muscle force is evaluated as stimulating the nerve vs the muscle: what do these data tell us?

13) CD68 is a marker for monocytes as well as macrophages.

14) Throughout multiple comparisons in the figure panels, is it necessary to include the bars and notation where there are no significant differences (i.e., eliminate those for "ns").

Reviewer 2 has the following overall critiques:

Despite the strengths of the study, some limitations are identified. Specifically, the impact of maresin 1 on macrophage phenotypes (M1/M2) could have been explored in more detail using histological or protein expression analysis. Moreover, additional data are needed to substantiate the claims about increased muscle regeneration. Lastly, the study does not address myofiber innervation, myofiber-type transitions, or motor unit remodeling.

Reviewer 2 raises the following specific issues for addressing:

The study also has some weaknesses that should be addressed prior to publication. The manuscript should be revised based on the following comments.

1. Please normalize the measured isometric torque by either dividing it by the weight of the TA muscle or the animal's body weight. This normalization is important because the torque value can vary with body mass, and normalizing the data to body mass will allow for more accurate comparisons between the experimental groups.

2. Can you provide the measurements of the biopsy mass that was removed to induce VML injury in each experimental group? This information is crucial for the reproducibility of the VML injury model and to enable comparison of the severity of the resulting injury.

3. The number of mice used for each experimental group is not mentioned in the methods section. Please mention the number of mice used for each group at each time point. Please also indicate if the injuries were unilateral or bilateral.

4. The reporting of sample sizes is not consistent. For instance, in Figure 1, it is n=4-5 per group, but in figure 2, it is n=5 mice per injury type and timepoint, and in Figure 4, it is n = 3-4 tissues per condition. Please edit for consistency. Is n=3 mice the same as n=3 tissues?

5. There are a few issues with the reporting of results that need to be addressed. Firstly, in the caption of figure 3F, the sample size is listed as n=7 mice per group, while in the Results section, a range of 7-11 mice per group is mentioned. This discrepancy needs to be reconciled to ensure accurate reporting of the sample size used in the study.

6. Secondly, there is some confusion regarding the statistical significance of the results. While the caption of figure 3F suggests a trend (implying p<0.1) for larger eMHC+ myofibers with Maresin 1 treatment, the graph shows an asterisk symbol suggesting p<0.05. It is recommended to double-check the data set to clarify the level of significance and ensure consistency in reporting.

7. Additionally, based on supplementary figure 3, it appears that there are more eMHC+ myofibers in the control group. Therefore, total counts for the eMHC+ myofibers in the entire section should be provided to better understand the distribution of these fibers among the groups.

8. Evidence for enhanced in vivo myogenesis is scarce, and eMHC+ myofiber data has inconsistencies in presentation (as mentioned above). Please provide additional data (e.g., percentage of myofibers with displaced nuclei, myofiber counts/cross-sectional area, protein expression of MRFs, increased Pax7+ satellite cell quantity) to better support the claim that maresin 1 is improving myogenesis.

9. Based on the evidence provided, it appears that maresin 1 is suppressing inflammation and fibrosis, which may be sparing more myofibers from undergoing necrosis in the surrounding musculature. Please discuss this possibility in the manuscript.

10. Figure 5 presents CD86, IL-1 β, CCR7 etc. as M2 macrophage associated markers, which is incorrect. These markers are indicative of M1 phenotype. Which specific markers were used to indicate the M2 phenotype?

*Reviewer #1 (Recommendations for the authors):*

1) The authors use 1mm vs 2mm punch biopsies to create "regenerative" vs "degenerative" injuries in the mouse tibialis anterior muscle. In a key reference (25), the threshold for successful regeneration following a biopsy of the mouse quadriceps muscle was defined as 3 mm; below this value (e.g., 2 mm) regeneration was found to be successful. Does the size of the injury for effective regeneration vary with the size of the muscle?

2) The authors introduce "dpi" to denote days post-injury. However, this acronym is not used consistently either in the text, figures, or figure captions (see comment 9 below).

3) The difference between legs in the size of the defect is described in the caption of Figure 2 but the difference is illustrated in Figure 1. Recommend revising the caption to Figure 1 accordingly. The notation used at the top of panel B is inconsistent.

4) It is not clear when (dpi) some of the data in the figures were obtained. A good example is panels C, E, and F in Figure 3, where time points are given for other panels but not these. Recommend perusal and revision throughout the text and figures (captions) for consistency. It is also not clear why key variables were not measured at respective time points (e.g., collagen deposition at 28 dpi when force was evaluated).

5) It is not clear which colors represent EdU incorporation or Wnt expression in Figure 4; for those not familiar with respective techniques, it would be helpful to identify these details. A specific concern is that the images in panels B and D are not convincing with respect to illustrating clear differences between 0 and 10 nM Maresin 1 treatment in B, or between Control and Knockdown in D.

6) An additional concern exemplified by the data in Panels C and E of Figure 4 is the low sample size (n=3), particularly given the variability in the data. There is no mention of tests for statistical power in the manuscript. It is recommended that such tests be applied for each of the variables upon which conclusions are based and the sample size adjusted accordingly to strengthen the conclusions presented to the reader.

7) In Figure 5, the UMAP plots for Vehicle and Maresin 1 are indistinguishable, making it difficult for the reader to identify the differences indicated in the text. Further, the text describing these data on p.5 indicates that respective conditions represent a pool of 3 tissues from 3 mice. It is not clear how statistics can be performed by comparing one pooled sample vs another pooled sample. It would also be helpful to clarify the timepoint (dpi) for these data in the figure caption.

8) It is remarkable that what is presented as a statistically significant difference for endothelial cells in Figure 5B is not addressed in the text. A recent study in (10.1186/s13395-023-00313-3) points to angiogenesis (i.e., endothelial cell proliferation) as an integral component of regeneration following a punch biopsy injury.

9) Panel A in Supplemental Figure 2 is confusing. There are more colors for respective clusters than are indicated on the Y-axis. Per comment 2), there are two different formats used here to identify time points, neither of which is dpi. Recommend revising all figures, captions, and text to use dpi for consistency. It may be helpful to define uninjured as 0 dpi.

10) Text in the Discussion associated with "at this timepoint" (p.6, paragraph 1, line 18) is confusing. Clarify intent.

11) Text in the Discussion associated with "did not vary significantly with larger injury sizes" (p.6, paragraph 2, line 8) is confusing. Clarify intent.

12) In Supplemental Figure 1, Isometric Force is presented as "Tetanus" throughout the range of stimulation frequency in panels A and B. Clarify what is going on at lower frequencies and which values are used in panels C, D, and E. It would also be helpful to clarify why muscle force is evaluated as stimulating the nerve vs the muscle: what do these data tell us?

13) CD68 is a marker for monocytes as well as macrophages.

14) Throughout multiple comparisons in the figure panels, is it necessary to include the bars and notation where there are no significant differences (i.e., eliminate those for "ns")?

*Reviewer #2 (Recommendations for the authors):*

The manuscript shows that large or non-healing VML injuries had an abundance of proinflammatory eicosanoids but lacked pro-resolving lipid mediators. The work also shows that treating VML injuries with Maresin 1, a pro-resolving lipid mediator derived from docosahexaenoic acid, reduced fibrotic tissue and neutrophil infiltration, ultimately resulting in increased muscle strength. The study also has some weaknesses that should be addressed prior to publication. The manuscript should be revised based on the following comments.

1. Please normalize the measured isometric torque by either dividing it by the weight of the TA muscle or the animal's body weight. This normalization is important because the torque value can vary with body mass, and normalizing the data to body mass will allow for more accurate comparisons between the experimental groups.

2. Can you provide the measurements of the biopsy mass that was removed to induce VML injury in each experimental group? This information is crucial for the reproducibility of the VML injury model and to enable comparison of the severity of the resulting injury.

3. The number of mice used for each experimental group is not mentioned in the methods section. Please mention the number of mice used for each group at each time point. Please also indicate if the injuries were unilateral or bilateral.

4. The reporting of sample sizes is not consistent. For instance, in Figure 1, it is n=4-5 per group, but in figure 2, it is n=5 mice per injury type and timepoint, and in Figure 4, it is n = 3-4 tissues per condition. Please edit for consistency. Is n=3 mice the same as n=3 tissues?

5. There are a few issues with the reporting of results that need to be addressed. Firstly, in the caption of figure 3F, the sample size is listed as n=7 mice per group, while in the Results section, a range of 7-11 mice per group is mentioned. This discrepancy needs to be reconciled to ensure accurate reporting of the sample size used in the study.

6. Secondly, there is some confusion regarding the statistical significance of the results. While the caption of figure 3F suggests a trend (implying p<0.1) for larger eMHC+ myofibers with Maresin 1 treatment, the graph shows an asterisk symbol suggesting p<0.05. It is recommended to double-check the data set to clarify the level of significance and ensure consistency in reporting.

7. Additionally, based on supplementary figure 3, it appears that there are more eMHC+ myofibers in the control group. Therefore, total counts for the eMHC+ myofibers in the entire section should be provided to better understand the distribution of these fibers among the groups.

8. Evidence for enhanced in vivo myogenesis is scarce, and eMHC+ myofiber data has inconsistencies in presentation (as mentioned above). Please provide additional data (e.g., percentage of myofibers with displaced nuclei, myofiber counts/cross-sectional area, protein expression of MRFs, increased Pax7+ satellite cell quantity) to better support the claim that maresin 1 is improving myogenesis.

9. Based on the evidence provided, it appears that maresin 1 is suppressing inflammation and fibrosis, which may be sparing more myofibers from undergoing necrosis in the surrounding musculature. Please discuss this possibility in the manuscript.

10. Figure 5 presents CD86, IL-1 β, CCR7 etc. as M2 macrophage associated markers, which is incorrect. These markers are indicative of M1 phenotype. Which specific markers were used to indicate the M2 phenotype?

---

## [Author Response]

Essential revisions:Reviewer (1) has the following general comments:Certain aspects of the data are overinterpreted; while some measures appear to have an adequate sample size to make sound conclusions, other measures are likely to lack sufficient statistical power given their variability. Presentation of the results would be strengthened by adhering to consistent terminology and labeling of figures throughout; specific examples are identified in recommendations to the authors. Several of the images used to illustrate differences between treatments are unconvincing because differences are not readily.

We agree with the reviewer and have scaled back some of the interpretation as well as clarified the sample sizes. We have also amended the text to maintain a consistent terminology.

Reviewer (2) raises the following specific issues that require addressing:1) The authors use 1mm vs 2mm punch biopsies to create "regenerative" vs "degenerative" injuries in the mouse tibialis anterior muscle. In a key reference (25), the threshold for successful regeneration following a biopsy of the mouse quadriceps muscle was defined as 3 mm; below this value (e.g., 2 mm) regeneration was found to be successful. Does the size of the injury for effective regeneration vary with the size of the muscle?

We agree with the reviewer and appreciate this comment. In the previous model (reference 25) and in our own results (reference 10), a 2mm defect applied to the quadriceps resulted in regeneration. Since the tibialis anterior muscle is smaller than the quadriceps, a 2mm defect to the tibialis anterior muscle resulted in approximately 30% of tissue removed. This is comparable to the amount of tissue removed in the quadriceps that resulted in a degenerative VML injury.

2) The authors introduce "dpi" to denote days post-injury. However, this acronym is not used consistently either in the text, figures, or figure captions (see comment 9 below).3) The difference between legs in the size of the defect is described in the caption of Figure 2 but the difference is illustrated in Figure 1. Recommend revising the caption to Figure 1 accordingly. The notation used at the top of panel B is inconsistent.

We have amended the figure captions and figures to address the inconsistency.

4) It is not clear when (dpi) some of the data in the figures were obtained. A good example is panels C, E, and F in Figure 3, where time points are given for other panels but not these. Recommend perusal and revision throughout the text and figures (captions) for consistency. It is also not clear why key variables were not measured at respective time points (e.g., collagen deposition at 28 dpi when force was evaluated).

We appreciate this comment from the reviewer and agree that adding in time points to some of the figures will aid in the interpretation of the dataset. We have revised the figures to ensure that time points from the associated measurements have been added as well as included these descriptions in the captions.

5) It is not clear which colors represent EdU incorporation or Wnt expression in Figure 4; for those not familiar with respective techniques, it would be helpful to identify these details. A specific concern is that the images in panels B and D are not convincing with respect to illustrating clear differences between 0 and 10 nM Maresin 1 treatment in B, or between Control and Knockdown in D.

Thank you for bringing this to our attention. We have replaced the EdU images to illustrate that a higher percentage of muscle stem cells treated with Maresin 1 incorporated EdU, in line with the quantitation. We have also further validated the mechanism of enhanced proliferation from Maresin 1 treatment by assaying cyclic AMP levels (new Figure 4C) and observed enhancements in cyclic AMP with Maresin 1 treatment, which is in line with our observations of increased proliferation.

6) An additional concern exemplified by the data in Panels C and E of Figure 4 is the low sample size (n=3), particularly given the variability in the data. There is no mention of tests for statistical power in the manuscript. It is recommended that such tests be applied for each of the variables upon which conclusions are based and the sample size adjusted accordingly to strengthen the conclusions presented to the reader.

We agree that the statistical power for this experiment was low and revamped Figure 4. First, we repeated experiments on the canonical Wnt fluorescent reporter in MuSCs (Pax7^CreERT2^ – Rosa26^-Tcf/Lef-LSL-H2B-GFP^), as well as performed additional experiments to understand how changes in canonical Wnt signaling are enacted in MuSCs with Maresin 1 treatment. First, we found that treatment of MuSCs with Maresin 1 did not change β catenin levels (new Figure 4D), suggesting that the enhancements in proliferation were not mediated through this signaling pathway. This result is consistent with previous observations that Maresin 1 did not significantly stimulate canonical Wnt in mesenchymal stem cells (Khedgikar et al., Bone 2022 or reference #33). We additionally performed culture of MuSCs that harbor the canonical Wnt fluorescent reporter with or without Wnt3a (a canonical Wnt ligand) and detected increases in the % of GFP+ MuSCs (new Supplemental Figure 5B). Power analyses were performed using data from pilot experiments and sample sizes were set such that a power of 80% was achieved. This is included at the end of the methods subsections for each experiment. The results show that the increase in proliferation of MuSCs by Maresin 1 is not mediated by canonical Wnt but through cyclic AMP.

7) In Figure 5, the UMAP plots for Vehicle and Maresin 1 are indistinguishable, making it difficult for the reader to identify the differences indicated in the text. Further, the text describing these data on p.5 indicates that respective conditions represent a pool of 3 tissues from 3 mice. It is not clear how statistics can be performed by comparing one pooled sample vs another pooled sample. It would also be helpful to clarify the timepoint (dpi) for these data in the figure caption.

We thank the reviewer for bringing this to our attention. We have rearranged the figure to enhance the clarity between treatments and added a notation for the day the samples were isolated. To increase the number of cells sequenced, sample pooling was needed. We have also added an additional sequencing replicate from each condition and increased the number of cells sequenced to enhance our statistical analysis.

8) It is remarkable that what is presented as a statistically significant difference for endothelial cells in Figure 5B is not addressed in the text. A recent study in (10.1186/s13395-023-00313-3) points to angiogenesis (i.e., endothelial cell proliferation) as an integral component of regeneration following a punch biopsy injury.

We agree on the importance of angiogenesis and the potential link that our findings might be pointing to. Although specific experiments are needed to further support this claim, which are beyond the scope of this manuscript, we have included the suggested reference and added several lines of interpretation into the Discussion section.

9) Panel A in Supplemental Figure 2 is confusing. There are more colors for respective clusters than are indicated on the Y-axis. Per comment 2), there are two different formats used here to identify time points, neither of which is dpi. Recommend revising all figures, captions, and text to use dpi for consistency. It may be helpful to define uninjured as 0 dpi.

We apologize for the confusion. The colors of the plots in Supplemental Figure 2A are the same as the colors corresponding to the clusters indicated in the heatmap in Figure 2B. The clusters were determined by DPGP clustering (not size of injury). As noted above, we have revised figures and used the dpi format for all figures and 0 dpi was used instead of uninjured.

10) Text in the Discussion associated with "at this timepoint" (p.6, paragraph 1, line 18) is confusing. Clarify intent.

We thank the reviewer and agree that this sentence is confusing. We have removed this sentence from the discussion.

11) Text in the Discussion associated with "did not vary significantly with larger injury sizes" (p.6, paragraph 2, line 8) is confusing. Clarify intent.

We thank the reviewer for the observation. We have revised this sentence and it now reads:

“While we observed that pro-resolving lipid mediators are expressed in VML injuries, the level at which these factors are detected did not proportionally increase with the heightened inflammation observed in 2mm injury compared to 1mm injury.”

12) In Supplemental Figure 1, Isometric Force is presented as "Tetanus" throughout the range of stimulation frequency in panels A and B. Clarify what is going on at lower frequencies and which values are used in panels C, D, and E. It would also be helpful to clarify why muscle force is evaluated as stimulating the nerve vs the muscle: what do these data tell us?

We appreciate this comment from the reviewer. We first changed the title of the y-axis in figures in Supplemental Figures 1A-1B to “Isometric Force (mN)”. We described the frequency at which tetanic force was measured in the “In situ functional testing” section of the Methods. This measurement typically was at 200 Hz and this has been added into the figure caption. It is common practice to evaluate nervous and muscle stimulation for functional testing (Dellorusso et al. Journal of Muscle Res. and Cell Motil. 22: 467–475, 2001) before and after injury as an indicator of functonal neurotranmission, i.e. lower force generation elicited with nerve stimulation than with direct muscle stimulation suggests that there are muscle fibers that cannot be activated by nerve stimulation, implicating a neurotransmission defect. It is unknown how innervation levels are restored after VML, and these results show comparable values for force output between neural and muscle stimulation suggesting that functional innervation was fully restored. Further experiments are needed to evaluate how motor endplates change after VML, which is beyond the scope of this manuscript.

13) CD68 is a marker for monocytes as well as macrophages.

We thank for the reviewer for this comment. We have amended the text to say monocytes and macrophages.

14) Throughout multiple comparisons in the figure panels, is it necessary to include the bars and notation where there are no significant differences (i.e., eliminate those for "ns").

We thank the reviewer for this comment and agree. Non-significant labels were removed from all figures.

Reviewer 2 has the following overall critiques:Despite the strengths of the study, some limitations are identified. Specifically, the impact of maresin 1 on macrophage phenotypes (M1/M2) could have been explored in more detail using histological or protein expression analysis. Moreover, additional data are needed to substantiate the claims about increased muscle regeneration. Lastly, the study does not address myofiber innervation, myofiber-type transitions, or motor unit remodeling.

We thank the reviewer for the suggestions and have performed a more in-depth exploration of macrophage phenotypes through additional scRNA-sequencing analysis. We have also included additional data describing how Maresin 1 impacts muscle stem cells through cyclic AMP. Respectfully, profiling myofiber innervation, motor unit remodeling and myofiber-type transitions are beyond the scope of this manuscript.

Reviewer 2 raises the following specific issues for addressing:The study also has some weaknesses that should be addressed prior to publication. The manuscript should be revised based on the following comments.1. Please normalize the measured isometric torque by either dividing it by the weight of the TA muscle or the animal's body weight. This normalization is important because the torque value can vary with body mass, and normalizing the data to body mass will allow for more accurate comparisons between the experimental groups.

Further details regarding the calculation of the specific force are detailed in the methods section. Briefly, specific force measurement uses muscle mass and muscle length (Lo) to normalize the data (we are not measuring torque). The absolute force and specific force which is normalized are both reported.

2. Can you provide the measurements of the biopsy mass that was removed to induce VML injury in each experimental group? This information is crucial for the reproducibility of the VML injury model and to enable comparison of the severity of the resulting injury.

The muscle mass that was removed for the 2mm defects was approximately 6-8mg, representing about 30% of TA muscle mass. We have added this to the methods section describing the injury.

3. The number of mice used for each experimental group is not mentioned in the methods section. Please mention the number of mice used for each group at each time point. Please also indicate if the injuries were unilateral or bilateral.

The sample sizes for each experimental group are included in each of the figure legends. Injuries that were bi-lateral have been labeled.

4. The reporting of sample sizes is not consistent. For instance, in Figure 1, it is n=4-5 per group, but in figure 2, it is n=5 mice per injury type and timepoint, and in Figure 4, it is n = 3-4 tissues per condition. Please edit for consistency. Is n=3 mice the same as n=3 tissues?

We have amended the figure captions to include both numbers of tissues and numbers of mice used for each experiment. The numbers of mice required for each experiment were different based on power analysis.

5. There are a few issues with the reporting of results that need to be addressed. Firstly, in the caption of figure 3F, the sample size is listed as n=7 mice per group, while in the Results section, a range of 7-11 mice per group is mentioned. This discrepancy needs to be reconciled to ensure accurate reporting of the sample size used in the study.

We thank the reviewer for bringing this to our attention and have corrected the sample sizes reported.

6. Secondly, there is some confusion regarding the statistical significance of the results. While the caption of figure 3F suggests a trend (implying p<0.1) for larger eMHC+ myofibers with Maresin 1 treatment, the graph shows an asterisk symbol suggesting p<0.05. It is recommended to double-check the data set to clarify the level of significance and ensure consistency in reporting.

We apologize for this confusion and have revised the figure caption to match the figure representation of statistical significance.

7. Additionally, based on supplementary figure 3, it appears that there are more eMHC+ myofibers in the control group. Therefore, total counts for the eMHC+ myofibers in the entire section should be provided to better understand the distribution of these fibers among the groups.

We thank the reviewer for the observation. We have revised the label and included both count and percentages in the revised supplemental figures. We have also included the count and percentage of centrally located nuclei in the revised supplemental figures.

8. Evidence for enhanced in vivo myogenesis is scarce, and eMHC+ myofiber data has inconsistencies in presentation (as mentioned above). Please provide additional data (e.g., percentage of myofibers with displaced nuclei, myofiber counts/cross-sectional area, protein expression of MRFs, increased Pax7+ satellite cell quantity) to better support the claim that maresin 1 is improving myogenesis.

We thank the reviewer for pointing this out. We have included data for centrally nucleated myofibers in Supplemental Figure 4C, and overall cross-sectional areas of eMyHC fibers. We also performed new experiments using a MuSC fluorescent reporter (Pax7^CreERT2^ – Rosa26^-TdTomato^) to further understand changes in muscle regeneration. These results are portrayed in Figure 3F-G and show a small improvement in the size of myofibers that contain TdTomato (and a trend of increased myofiber size shown in Supplemental Information Figure 4). These results agree with the eMyHC data showing small improvements in muscle cross-sectional area as a result of Maresin 1 treatment. Last, we measured gene expression of several MRFs and other regeneration associated genes such as Tmem8c/ Myomaker, Titin (Ttn) and myogenin are plotted in the new Figure 5F, which shows differentially expressed genes in muscle stem cells measured by scRNAseq.

9. Based on the evidence provided, it appears that maresin 1 is suppressing inflammation and fibrosis, which may be sparing more myofibers from undergoing necrosis in the surrounding musculature. Please discuss this possibility in the manuscript.

We thank the reviewer for this comment and very much agree with this statement. We have added this into the discussion.

10. Figure 5 presents CD86, IL-1 β, CCR7 etc. as M2 macrophage associated markers, which is incorrect. These markers are indicative of M1 phenotype. Which specific markers were used to indicate the M2 phenotype?

We thank the reviewer for pointing this out, and we agree that a more comprehensive analysis of macrophage subsets was needed. CD86 and Il1b are poor markers since M2b macrophages and M1 macrophages express both. Il1r2 is primarily expressed by M2a macrophages. We have generated additional scRNA-Seq data and performed a more comprehensive analysis of macrophage phenotypes post Maresin treatment.